# Mice deficient of *Myc* super-enhancer region reveal differential control mechanism between normal and pathological growth

Kashyap Dave[1†], Inderpreet Sur[1†], Jian Yan[1], Jilin Zhang[1], Eevi Kaasinen[1], Fan Zhong[1], Leander Blaas[2], Xiaoze Li[3], Shabnam Kharazi[3], Charlotte Gustafsson[3], Ayla De Paepe[3], Robert Månsson[3], Jussi Taipale[1,4]*

[1]Division of Functional Genomics and Systems Biology, Department of Medical Biochemistry and Biophysics, Karolinska Institutet, Stockholm, Sweden; [2]Department of Biosciences and Nutrition, Karolinska Institutet, Stockholm, Sweden; [3]Center for Hematology and Regenerative Medicine, Karolinska Institutet, Stockholm, Sweden; [4]Genome-Scale Biology Program, University of Helsinki, Helsinki, Finland

*For correspondence: jussi. taipale@ki.se

[†]These authors contributed equally to this work

Competing interests: The authors declare that no competing interests exist.

**Abstract** The gene desert upstream of the *MYC* oncogene on chromosome 8q24 contains susceptibility loci for several major forms of human cancer. The region shows high conservation between human and mouse and contains multiple *MYC* enhancers that are activated in tumor cells. However, the role of this region in normal development has not been addressed. Here we show that a 538 kb deletion of the entire *MYC* upstream super-enhancer region in mice results in 50% to 80% decrease in *Myc* expression in multiple tissues. The mice are viable and show no overt phenotype. However, they are resistant to tumorigenesis, and most normal cells isolated from them grow slowly in culture. These results reveal that only cells whose MYC activity is increased by serum or oncogenic driver mutations depend on the 8q24 super-enhancer region, and indicate that targeting the activity of this element is a promising strategy of cancer chemoprevention and therapy.

## Introduction

Deregulated expression of the *MYC* oncogene is associated with many cancer types (Reviewed in *Albihn et al., 2010*; *Dang, 2012*; *Evan, 2012*). MYC acts primarily as a transcriptional activator that increases expression of many genes required for RNA and protein synthesis above the level that is required in resting cells. In cancer cells, aberrantly elevated levels of MYC drive global amplification of transcription rates, providing the cells with necessary resources for rapid proliferation (see, for example *Brown et al., 2008*; *van Riggelen et al., 2010*; *Ji et al., 2011*; *Lin et al., 2012*; *Sabò et al., 2014*; *Walz et al., 2014*).

Transcription of the *MYC* gene is regulated by a diverse array of regulatory elements located both upstream and downstream of the *MYC* transcription start site (TSS). Variants in the *MYC* upstream region contribute to inherited susceptibility to most major forms of human cancer, and account for a very large number of cancer cases at the population level (*Amundadottir et al., 2006*; *Gudmundsson et al., 2007*; *Yeager et al., 2007*; *Al Olama et al., 2009*; *Yeager et al., 2009*). For example, the polymorphism rs6983267 linked to colorectal (*Tomlinson et al., 2007*) and prostate (*Yeager et al., 2007*) cancers contributes more to cancer morbidity and mortality than any other

**eLife digest** Our cells each contain close to 20,000 genes, which provide the instructions needed to build our bodies and keep us alive. At any one time in the life of the cell, only some of these genes are active. The activity of each gene is constantly regulated to allow the cell to respond to changes in its environment. Enhancers are sections of DNA, outside of the genes, that act as molecular switches controlling the activity of genes. A gene can have many such enhancers; each enhancer is linked to a particular set of signals and having multiple enhancers allows the same gene to be activated by different signals in different tissues in the body.

Changes to enhancers can have serious consequences. By altering the activity of genes, an enhancer can have widespread effects on the health and behavior of a cell, including transforming it from healthy to cancerous. The small differences in enhancers also make some people more susceptible to cancers than others. If we can identify enhancers whose activity is commonly altered in cancers, it could be possible to target them through treatment. Yet, it is not clear whether targeting enhancers in this way could be effectively used to treat cancer without damaging healthy cells.

Now, Dave, Sur et al. have examined a large enhancer region with known links to several different cancers – including prostate, breast and colon cancers – to uncover whether it also plays a critical role in healthy cells and if it could be safely targeted for treatment. The region has multiple enhancers for a cancer-linked gene called *MYC* and is implicated in many cancer-associated deaths every year. This particular enhancer region is found in both humans and mice, which share many genes in common. Using genetic engineering, Dave, Sur et al. removed this enhancer region from the genetic information of a group of mice. The experiment showed that mice without the enhancer region were completely healthy. Also, when tested for cancer development, these mice were much less susceptible to several major types of cancer.

This investigation reveals that it may be possible to create drugs to shut down or inhibit certain enhancers to prevent or treat cancer without damaging healthy cells. However, this is currently just one example in mice under laboratory conditions. Further research is needed to determine if a similar approach can be developed to treat patients in the clinic.

known inherited variant or mutation, including the inherited mutations in classic tumor suppressors such as *RB*, *TP53* and *APC*. Through computational and experimental analyses, we and others have shown that the risk allele G of rs6983267 creates a strong binding site for the colorectal-cancer associated transcription factor Tcf7l2 (*Pomerantz et al., 2009*; *Tuupanen et al., 2009*). This binding site is located within the *Myc-335* enhancer element that is dispensable for mouse viability, but required for efficient Tcf7l2-driven intestinal tumorigenesis (*Sur et al., 2012b*).

More recently, another enhancer element, located 1.47 Mb downstream of *Myc* was shown to be required for formation of acute lymphoblastic leukemia (ALL) in mice (*Herranz et al., 2014*). However, in contrast to the *Myc-335* element, this element is also required for normal T-cell development. Thus, the mechanism by which individual *Myc* enhancer elements contribute to normal development and tumorigenesis is still unclear.

Several studies have shown that the 8q24 region contains a large number of additional enhancer elements (see, for example [*Hallikas et al., 2006*; *Ahmadiyeh et al., 2010*; *Yan et al., 2013*; *Yao et al., 2014*]) and super-enhancers that are active in many different types of human cancer (*Hnisz et al., 2013*; *Lovén et al., 2013*; *Zhou et al., 2015*). The *MYC*-associated super-enhancers are activated during the process of tumorigenesis (*Hnisz et al., 2013*), and downregulation of super-enhancer activity leads to selective inhibition of *MYC* expression (*Lovén et al., 2013*). Thus, *MYC*-associated super-enhancer activity is required for tumorigenesis, but the role of these elements in normal tissue morphogenesis and homeostasis has been unclear.

To address this problem, we have in this work generated multiple mouse strains deficient of regulatory elements upstream of the *Myc* promoter. Since this region contains multiple tumor type and tissue -specific enhancers and super-enhancers, for the sake of clarity we refer to the deleted region here as the 'super-enhancer region'. By analysis of the mice, we found that the entire super-enhancer

region conferring multi-cancer susceptibility contributes to MYC expression *in vivo*, yet is not required for mouse embryonic development and viability. However, this region is required for the growth of normal cells in culture and cancer cells *in vivo*. As cultured cells are exposed to serum, which is a signal of tissue damage, this finding suggests that tumor cells and cells responding to damage signals share regulatory mechanisms that are dispensable for normal physiological growth control.

## Results

### Functional mapping of the super-enhancer region upstream of *Myc*

To dissect functional significance of the 8q24 region during normal development, we generated series of *Myc* alleles in mice using homologous recombination in ES cells. These include the *Myc-335* enhancer deletion allele we have described previously (*Sur et al., 2012b*), and deletions of two additional conserved enhancer elements, *Myc-196* and *Myc-540*, both of which are active in mouse intestine and colorectal cancer cells. In addition, we generated a point mutation that inactivates a conserved CCCTC-Binding factor (CTCF) site 2 kb upstream of the *Myc* TSS. This site has previously been reported to be required for *MYC* expression (*Gombert and Krumm, 2009*), and to have insulator activity (*Gombert et al., 2003*) (*Figure 1a*). Each allele contained loxP site(s) in the same orientation to allow conditional knockouts of the enhancers, and to facilitate generation of large deletions and duplications by interallelic recombination (*Wu et al., 2007*). All alleles were bred to homozygosity, and resulted in generation of viable mice. Expression of *Myc* in the colon of *Myc-196$^{-/-}$* and *Myc-540$^{-/-}$* mice was not markedly altered, suggesting that these elements have little effect on regulation of *Myc* in the intestine under normal laboratory conditions (*Figure 1b*). *Myc* expression level was also normal in *Myc-CTCF$^{mut/mut}$* mouse colon despite loss of CTCF and cohesin (Rad21) binding to the region proximal to the *Myc* promoter (*Figure 1c*).

### Mice lacking the *Myc* super-enhancer region are viable and fertile

As the individual mutations and deletions had limited effect, we next decided to generate two large deletions in the *Myc* locus using interallelic recombination between the *Myc-CTCF$^{mut}$* loxP site and the loxP sites at *Myc-335$^-$* and *Myc-540$^-$*, yielding deletions of 365 kb (GRCm38/mm10 chr15:61618287–61983375) and 538 kb (chr15:61445326–61983375), respectively (*Figure 2a*). The resulting alleles, *Myc$^{\triangle 2\text{-}367}$* and *Myc$^{\triangle 2\text{-}540}$*, were then segregated out from the corresponding duplications, and bred to homozygosity. Given the very large regions that were deleted (*Figure 2b*), we expected to see a strong phenotype. However, no overt phenotype was identified in the *Myc$^{\triangle 2\text{-}367/\triangle 2\text{-}367}$* mice. The mice were born at the expected mendelian ratio, and both males and females were viable and fertile. Analysis of *Myc* expression, however, revealed a strong decrease in *Myc* expression in the colon and ileum of the mice (not shown).

The larger deletion, *Myc$^{\triangle 2\text{-}540}$*, could also be bred to homozygosity, and both males and females were viable. Given that the entire *Myc* regulatory region spans more than 2 Mb of DNA and is located on both sides of the *Myc* coding region (*Rosenbloom et al., 2013*; *Sloan et al., 2016*), the deletion is not expected to be equivalent to deletion of the *Myc* gene itself. Still, the viability of the mice is striking, since the region deleted contains regions linked to risk for myeloma, chronic lymphocytic leukemia and pancreatic, thyroid, bladder, prostate, breast, and colon cancers (*Chung and Chanock, 2011*; *Sahasrabudhe et al., 2015*; *Mitchell et al., 2016*; *Zhang et al., 2016*). To characterize the mice further, we analyzed histology and MYC expression in the tissues where these tumors originate from. This analysis revealed normal morphology of mammary gland, spleen, bladder, prostate and colon in *Myc$^{\triangle 2\text{-}540/\triangle 2\text{-}540}$* mice (*Figure 2c*).

### Loss of the super-enhancer region leads to tissue-specific changes in *Myc* expression

Although the *Myc$^{\triangle 2\text{-}540/\triangle 2\text{-}540}$* mice exhibited a normal phenotype, *Myc* expression was altered in a tissue-specific manner in these mice. This is expected since this region contains individual tissue specific regulatory elements. The expression of *Myc* was strongly decreased in colon, small intestine and prostate of these mice (*Figure 3a* and not shown). Immunohistochemical analysis of MYC expression in intestine revealed strong decrease of nuclear staining, and loss of MYC expression

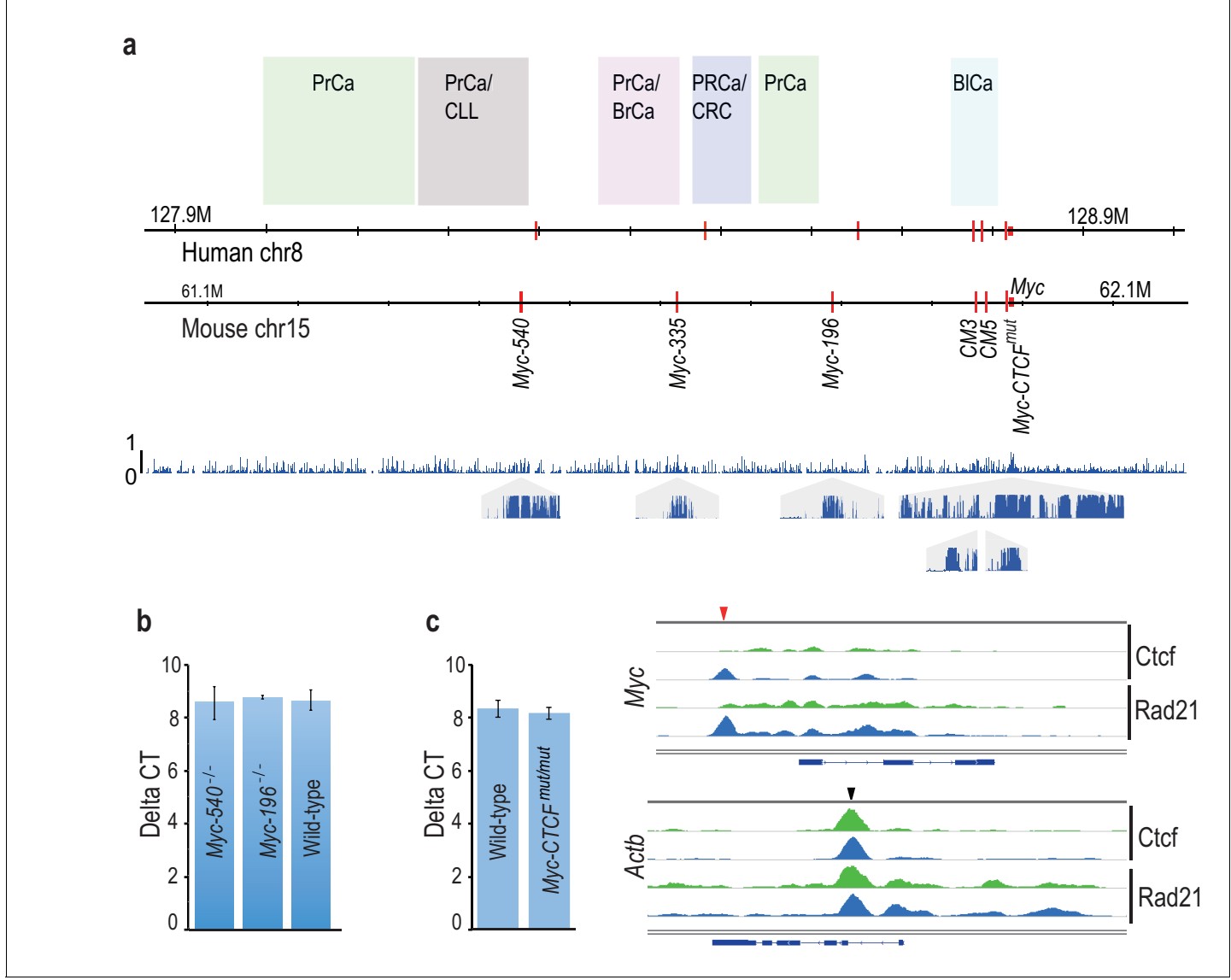

**Figure 1.** Cancer susceptibility region upstream of *Myc* contains several conserved enhancer elements that are dispensable for normal mouse development and MYC expression. (a) Comparison of *Myc* locus between human and mouse. The susceptibility regions for prostate cancer (PrCa), chronic lymphocytic leukemia (CLL), breast cancer (BrCa), colorectal cancer (CRC) and bladder cancer (BlCa) are marked. Red vertical lines mark the location of the Tcf7l2-binding CRC *Myc* enhancers in the two species. The lower panel shows the regional conservation probability predicted by PhastCons (hg19 assembly, UCSC) with non-overlapping sliding windows for the whole region and each enhancer locus with a size of 500 bp and 10 bp, respectively. (b) Deletion of *Myc-196* and *Myc-540* enhancer elements does not affect *Myc* expression in the colon as determined by qPCR analysis (*Myc-196*$^{-/-}$ n = 2, *Myc-540*$^{-/-}$ n = 3 and wild-type n = 5). See *Figure 1—source data 1* for details. (c) Mutation of the *Myc-CTCF* site causes loss of CTCF and Rad21 binding at the *Myc* locus (top panel). Binding of CTCF and Rad21 at a control *Actb* locus is not affected. Red and black arrowheads denote binding sites at *Myc* and *Actb* loci, respectively; green: *Myc-CTCF*$^{mut/mut}$, blue: wild-type. The gene body for *Myc* and *Actb* is shown below the respective panels. The qPCR analysis reveals that despite loss of CTCF/cohesin binding, the expression of *Myc* mRNA is not altered in the colon (for qPCR, *Myc-CTCF*$^{mut/mut}$ n = 4, wild-type n = 3). See *Figure 1—source data 1* for details. Error bars denote one standard deviation.

The following source data is available for figure 1:

**Source data 1.** *Myc* transcript levels in wild-type and mutant mice in *Figure 1b-c*.

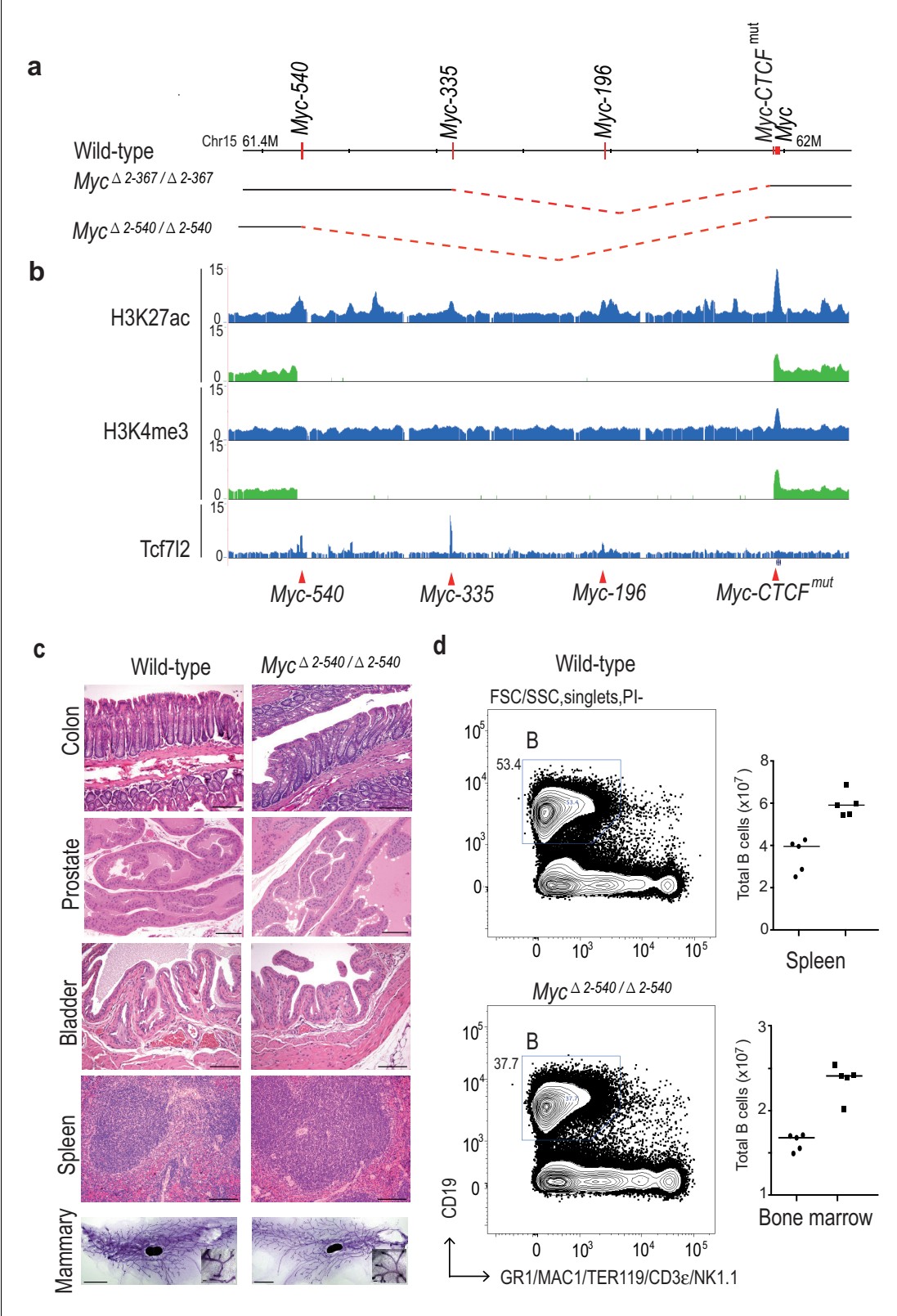

**Figure 2.** Deletion of the 8q24 super-enhancer region is well tolerated during normal development and homeostasis. (a) Schematic representation of the 365 kb and 538 kb deletions. (b) $Myc^{\triangle 2-540/\triangle 2-540}$ deletion removes a region containing several active enhancer elements upstream of $Myc$ as shown by ChIP-seq analysis of histone H3 lysine 27 acetylation (H3K27ac) and lysine four trimethylation (H3K4me3). The deletion also removes several Tcf7l2 ChIP-seq peaks. Signal from $Myc^{\triangle 2-540/\triangle 2-540}$ and wild-type mice are shown in green and blue, respectively. Red arrowheads and horizontal lines mark

*Figure 2 continued on next page*

*Figure 2 continued*

the different enhancer positions. (c) Haematoxylin/ Eosin stained sections of spleen, bladder, prostate, colon (Bar = 100 µm) and Carmine Alum stained whole mounts of mammary glands, Bars = 3 mm, 100 µm (inset) showing normal development and homeostasis of different organs in *Myc*$^{\triangle 2-540/\triangle 2-540}$ mice. (d) *Myc*$^{\triangle 2-540/\triangle 2-540}$ mice have a reduced number of B-cells compared to the wild-types. Left panel: FACS plots of a representative *Myc*$^{\triangle 2-540/\triangle 2-540}$ and wild-type mouse spleen showing B-cell (B) population. Right panel: Scatter dot plot of total number of B cells in the spleen and bone marrow of wild-type (squares), *n* = 5 and *Myc*$^{\triangle 2-540/\triangle 2-540}$ (filled circles), *n* = 5. Each point represents individual mouse. Line represents the median. See *Figure 2—source data 1* for details. The number of CD4$^+$ and CD8$^+$ T-cells is not affected by the deletion (see *Figure 2—figure supplement 1* and appendix 1).

The following source data and figure supplements are available for figure 2:

**Source data 1.** B cell numbers in the wild-type and *Myc*$^{\triangle 2-540/\triangle 2-540}$ mice in *Figure 2d*.
**Figure supplement 1.** The loss of the *Myc* super-enhancer region results in a decrease in the number of B-cells, but no major defects in hematopoiesis.
**Figure supplement 1—source data 1.** B and T-cell populations in the wild-type and *Myc*$^{\triangle 2-540/\triangle 2-540}$ mice in *Figure 2—figure supplement 1a*.

from the transit amplifying cell compartment. However, expression of MYC was still detected at the base of the crypt in the region where the intestinal stem cells are known to reside (*Figure 3b*). These results are consistent with the role of the deleted region in tumorigenesis of colon and prostate. To analyze the effect of decreased MYC expression on the proliferation in the transit amplifying

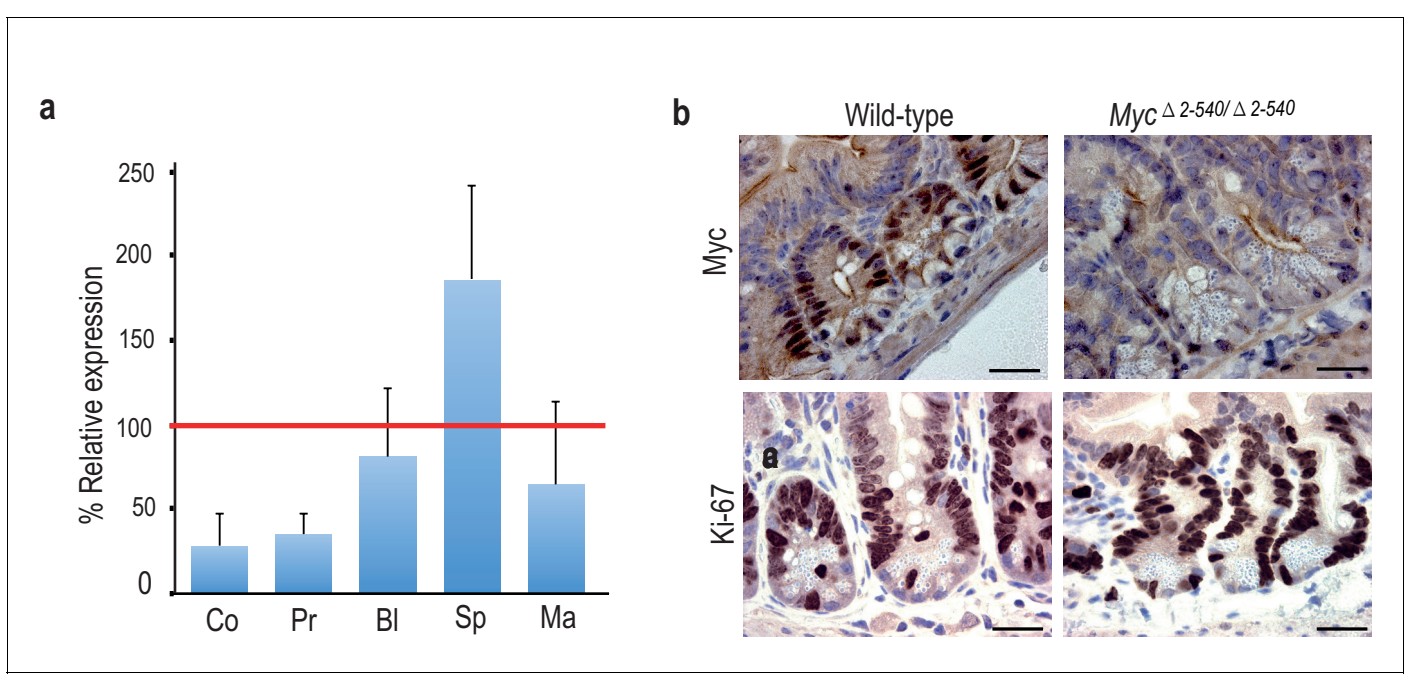

**Figure 3.** Tissue-specific effect of *Myc*$^{\triangle 2-540/\triangle 2-540}$ deletion on MYC expression. (a) qPCR data showing the percentage of *Myc* expression in *Myc*$^{\triangle 2-540/\triangle 2-540}$ relative to the wild-type in colon (Co) *n* = 4, prostate (Pr) *n* = 2, bladder (Bl) *n* = 5, spleen (Sp) *n* = 4 and mammary gland (Ma) *n* = 3. Red line marks the expression level (100%) in wild-type mice. Error bars indicate one standard deviation. See *Figure 3—source data 1* for details. (b) Immunohistochemistry shows reduced expression of MYC (*n* = 3 for each genotype) protein in intestinal crypts of *Myc*$^{\triangle 2-540/\triangle 2-540}$ mice without any significant effect on proliferation as indicated by Ki-67 (*n* = 2 for each genotype) immunostaining, Bar = 10 µm. Brown: IHC staining, Blue: Haematoxylin staining.
The following source data is available for figure 3:

**Source data 1.** Myc transcript levels in *Myc*$^{\triangle 2-540/\triangle 2-540}$ mice relative to the wild-types in *Figure 3a*.

compartment, we performed immunohistochemistry (IHC) for the proliferation marker Ki-67. Both the wild-type and $Myc^{\triangle 2-540/\triangle 2-540}$ had similar proliferative activity in the intestinal crypts (*Figure 3b*).

In contrast to colon and prostate, *Myc* expression was not markedly affected in the bladder, and was elevated in the spleen (*Figure 3a*). To analyze the cellular composition of the spleen, we performed flow cytometric analysis of markers for hematopoietic stem cells and lymphoid lineage cells. $Myc^{\triangle 2-540/\triangle 2-540}$ mice had a near normal hematopoietic compartment (*Figure 2d*). The only observed difference was a small reduction of B cells in the $Myc^{\triangle 2-540/\triangle 2-540}$ mice compared to the wild-type mice both in the spleen and the bone marrow. In contrast to the decrease in B-cells, the T cell numbers were not affected by the deletion (*Figure 2—figure supplement 1a*). This finding is consistent with the published data that regulatory elements controlling T-cell development and T-cell acute lymphoblastic leukemia are located 1.47 Mb downstream of the *Myc* ORF (*Herranz et al., 2014*).

To identify regulatory elements that could explain the effect in B-cells, we performed ChIP-seq analysis of chromatin from LSK-Flt3$^{neg}$ hematopoietic stem cells and mature B-cells isolated from wild-type mice. This analysis identified two B-cell specific regulatory elements. The *Myc 2–540* deletion results in loss of one of the elements, and moves the other element very close to the *Myc* TSS (*Figure 2—figure supplement 1b*). Although the exact regulatory mechanism is not clear and requires further study, the above data is consistent with a role of the super-enhancer region in development of chronic lymphocytic leukemia, which is primarily a B-cell malignancy. However, the decrease in B-cell number does not affect viability, and the $Myc^{\triangle 2-540/\triangle 2-540}$ mice are healthy and do not display an immune-deficient phenotype under normal 'clean' mouse housing conditions in the absence of known pathogenic microorganisms.

To compare the role of the 8q24 super-enhancer region in growth of cells *in vivo* and in cell culture, we isolated fibroblasts from the skin of adult $Myc^{\triangle 2-540/\triangle 2-540}$ and wild-type mice. Based on presence of active histone marks, and undermethylation of focal elements, the super-enhancer region is active in fibroblasts from both humans and mice (*Figure 4a* and *Figure 4—figure supplement 1*). However, the resident fibroblasts in the skin of $Myc^{\triangle 2-540/\triangle 2-540}$ mice appeared normal as judged by Vimentin expression (*Figure 4b*). Ki-67 staining (IHC) of skin sections showed comparable proliferation levels in wild-type and $Myc^{\triangle 2-540/\triangle 2-540}$ mice (*Figure 4b*). In contrast, most lines of fibroblasts (6 out of 7) isolated from $Myc^{\triangle 2-540/\triangle 2-540}$ mice grew slower in culture compared to fibroblasts isolated from wild-type mice (*Figure 4c*; p-value=0.0256, Mann-Whitney one tailed test).

## Deletion of the *Myc* super-enhancer region affects MYC target gene expression only under culture conditions

To understand the mechanism by which the deletion of the 8q24 super-enhancer region has a differential effect on growth during normal tissue homeostasis and growth under culture conditions, we subjected both the mouse tissues and cultured cells to RNA-seq analysis. Analysis of mouse tissues confirmed the changes in *Myc* expression observed by qPCR (*Figure 5a* and *Figure 5—figure supplement 1*). Surprisingly, despite more than 80% decrease of *Myc* expression in the colon, very few genes were downregulated in the tissues, and none of the significantly altered genes were known MYC targets (*Supplementary file 1*). These results suggest that expression of canonical MYC target genes is not sensitive to decreases in MYC protein level during normal tissue homeostasis. In contrast to the *in vivo* situation, where *Myc* is downregulated but key target genes are not affected, in cultured $Myc^{\triangle 2-540/\triangle 2-540}$ fibroblasts that grew slowly in culture, the downregulation of *Myc* lead to a loss of expression of key target genes that drive cell growth and division. Upstream regulator analysis performed using Ingenuity Pathway Analysis revealed that the highest-ranked potential regulator for the identified gene set was MYC (*Figure 5b*).

Measured by FPKM values, the cultured wild-type fibroblasts had higher *Myc* mRNA levels than normal tissues, whereas the cultured null fibroblasts had *Myc* levels that were comparable to or lower than those of normal wild-type tissues. The elevated *Myc* levels in cultured cells are caused by serum stimulation, as *Myc* mRNA levels are low in serum-starved fibroblasts and strongly induced by serum (Ref. [*Dean et al., 1986*] and our unpublished data). These results indicate that the 8q24 super-enhancer region is dispensable for normal tissue homeostasis under conditions where MYC activity is relatively low. However, the region is required for induction of MYC activity to levels that

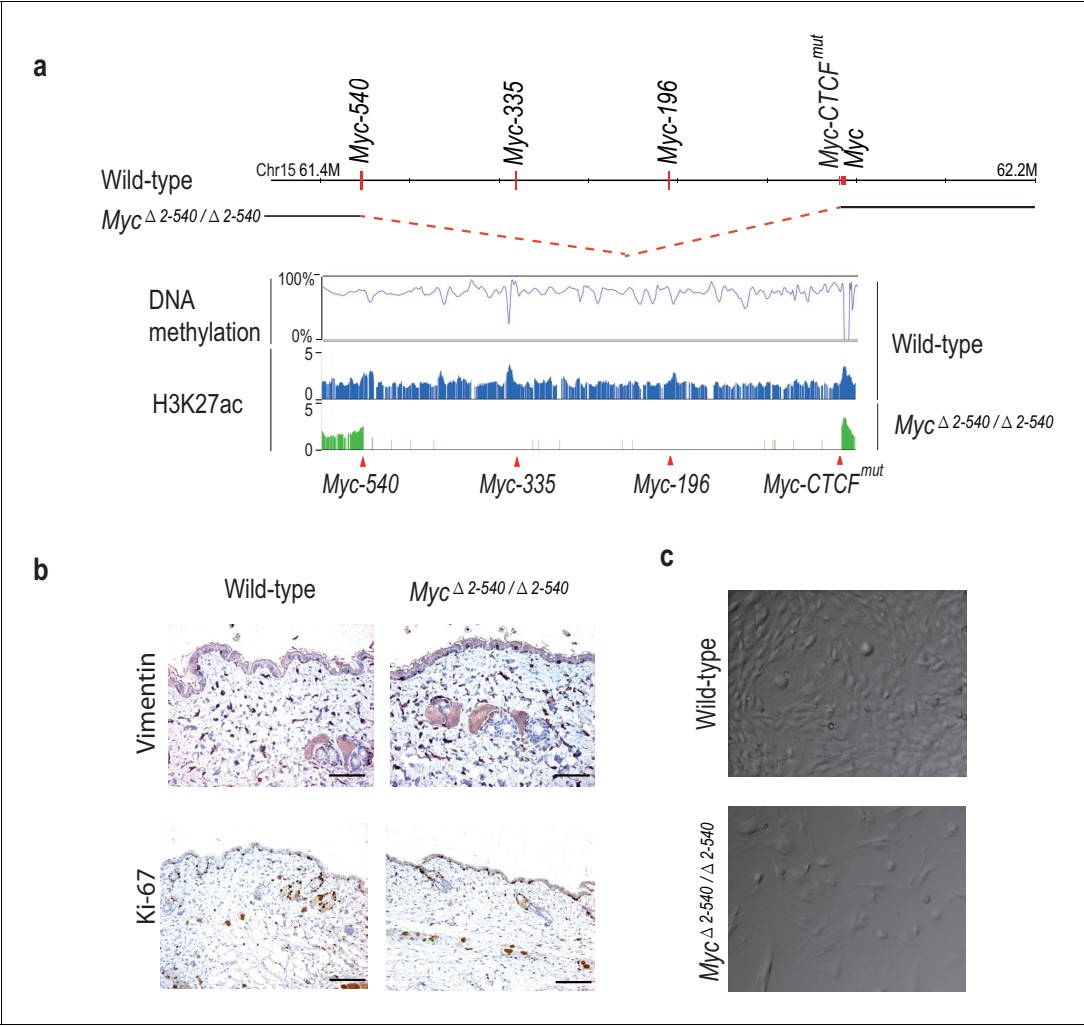

**Figure 4.** $Myc^{\triangle 2–540/\triangle 2–540}$ deletion results in a proliferation defect of adult skin fibroblasts cultured *in vitro*. (**a**) The super-enhancer region deleted in the $Myc^{\triangle 2–540/\triangle 2–540}$ has under methylated DNA as determined through bisulfite sequencing of the wild-type fibroblasts grown in culture. H3K27ac ChIP-seq shows the presence of active enhancer marks within this region in the wild-type fibroblasts whereas the $Myc^{\triangle 2–540/\triangle 2–540}$ fibroblasts show a complete absence of the super-enhancer region. The *Myc* super-enhancer region is also active in human fibroblasts (see *Figure 4—figure supplement 1*). (**b**) Normal morphology and proliferation of resident fibroblasts in the mouse skin as determined by Vimentin and Ki-67 IHC staining respectively in both the wild-type (n = 3) and $Myc^{\triangle 2–540/\triangle 2–540}$ mice (n = 3), Bar = 50 μm. Brown: IHC staining, Blue: Haematoxylin staining (**c**) Representative phase contrast images of wild-type and $Myc^{\triangle 2–540/\triangle 2–540}$ primary fibroblasts showing growth defect of $Myc^{\triangle 2–540/\triangle 2–540}$ fibroblasts in culture.

The following figure supplement is available for figure 4:

**Figure supplement 1.** The *Myc* super-enhancer region is also active in human fibroblasts.

are high enough to drive the expression of MYC target genes above their basal levels during pathological growth.

## The *Myc* super-enhancer region is required for tumorigenesis in mice

We have shown earlier that deletion of a 1.7 kb *Myc-335* enhancer sequence located at the 8q24 super-enhancer region is required for intestinal tumorigenesis in mice (*Sur et al., 2012b*). As the super-enhancer region deleted in $Myc^{\triangle 2–540/\triangle 2–540}$ mice carries risk also for other cancer types, including breast and bladder cancer, we tested the susceptibility of the $Myc^{\triangle 2–540/\triangle 2–540}$ mice to carcinogen induced bladder and mammary tumorigenesis. The $Myc^{\triangle 2–540/\triangle 2–540}$ mice were not resistant to N-Butyl-N(4-hydroxybutyl) nitrosamine (BBN) induced bladder tumors. Both wild-type (n = 8)

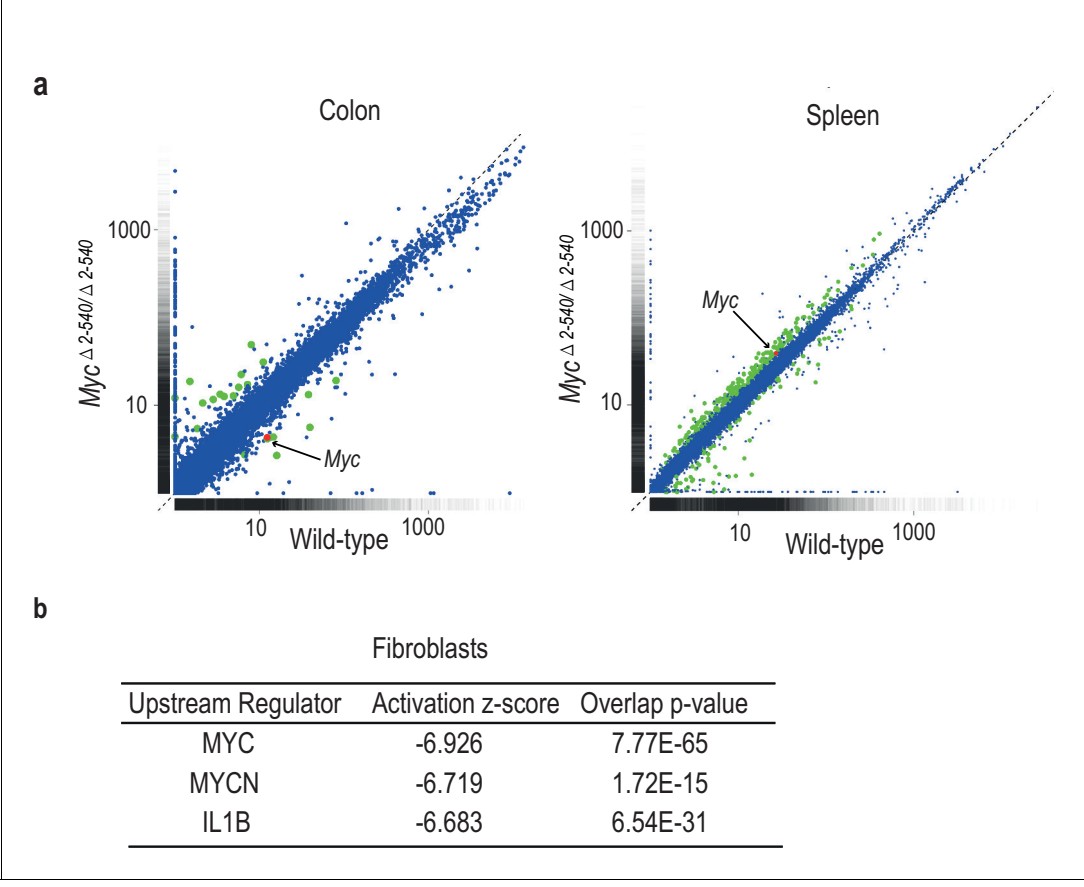

**Figure 5.** Differential effect of $Myc^{\triangle 2-540/\triangle 2-540}$ deletion on MYC target gene expression. (a) Scatter plot comparing the average Fragments per kilobase of exons per million fragments mapped (FPKM) values of gene transcripts in colon and spleen of wild-type (n = 4) and $Myc^{\triangle 2-540/\triangle 2-540}$ (n = 4) mice. Genes showing significant (q < 0.05) differential expression are marked in red (*Myc*) or green (other genes). For median FPKM values of gene transcripts see *Figure 5—figure supplement 1* (b) Upstream regulator analysis of RNA-seq data shows that the highest ranked potential regulator affected in the slow growing $Myc^{\triangle 2-540/\triangle 2-540}$ fibroblasts is MYC. The activation z-scores are to infer the activation states of predicted upstream regulators. The overlap *p*-values were calculated from all the regulator-targeted differential expression genes using Fisher's Exact Test. Two independent $Myc^{\triangle 2-540/\triangle 2-540}$ fibroblasts lines were analysed to confirm the downregulation of *Myc*. Ingenuity pathway analysis performed on one of these is shown.

The following figure supplement is available for figure 5:

**Figure supplement 1.** Scatter plot comparing the median of FPKM values of gene transcripts in colon of wild-type (n = 4) and $Myc^{\triangle 2-540/\triangle 2-540}$ (n = 4) mice.

and $Myc^{\triangle 2-540/\triangle 2-540}$ (n = 8) mice developed urothelial changes ranging from hyperplasia to high grade invasive urothelial carcinoma after 5 months of BBN treatment. In contrast, comparison of median tumor-free survival times of wild-type and $Myc^{\triangle 2-540/\triangle 2-540}$ mice exposed to mammary-tumor inducing dimethylbenz[a]anthracene/ medroxypregesterone (DMBA/MPA) regimen revealed that the $Myc^{\triangle 2-540/\triangle 2-540}$ mice were partially resistant to mammary tumorigenesis (*Figure 6a*). The median tumor-free survival time for the wild-type and $Myc^{\triangle 2-540/\triangle 2-540}$ mice was 88 and >120 days, respectively. Although we cannot pinpoint the specific regions that contribute to breast tumorigenesis by analysis of the $Myc^{\triangle 2-540/\triangle 2-540}$ mice, our work is consistent with the presence of a breast cancer susceptibility locus in humans at a region syntenic to the deletion. The region is distinct from the colon cancer susceptibility locus that harbors *Myc-335*.

To determine whether additional elements outside of the *Myc-335* region are playing a role in tumorigenesis, we crossed the $Myc^{\triangle 2-540/\triangle 2-540}$ mice with the $Apc^{min}$ mouse that is susceptible to intestinal tumors. The $Myc^{\triangle 2-540/\triangle 2-540}$ mice had fewer polyps than the $Myc\text{-}335^{-/-}$ mice in the

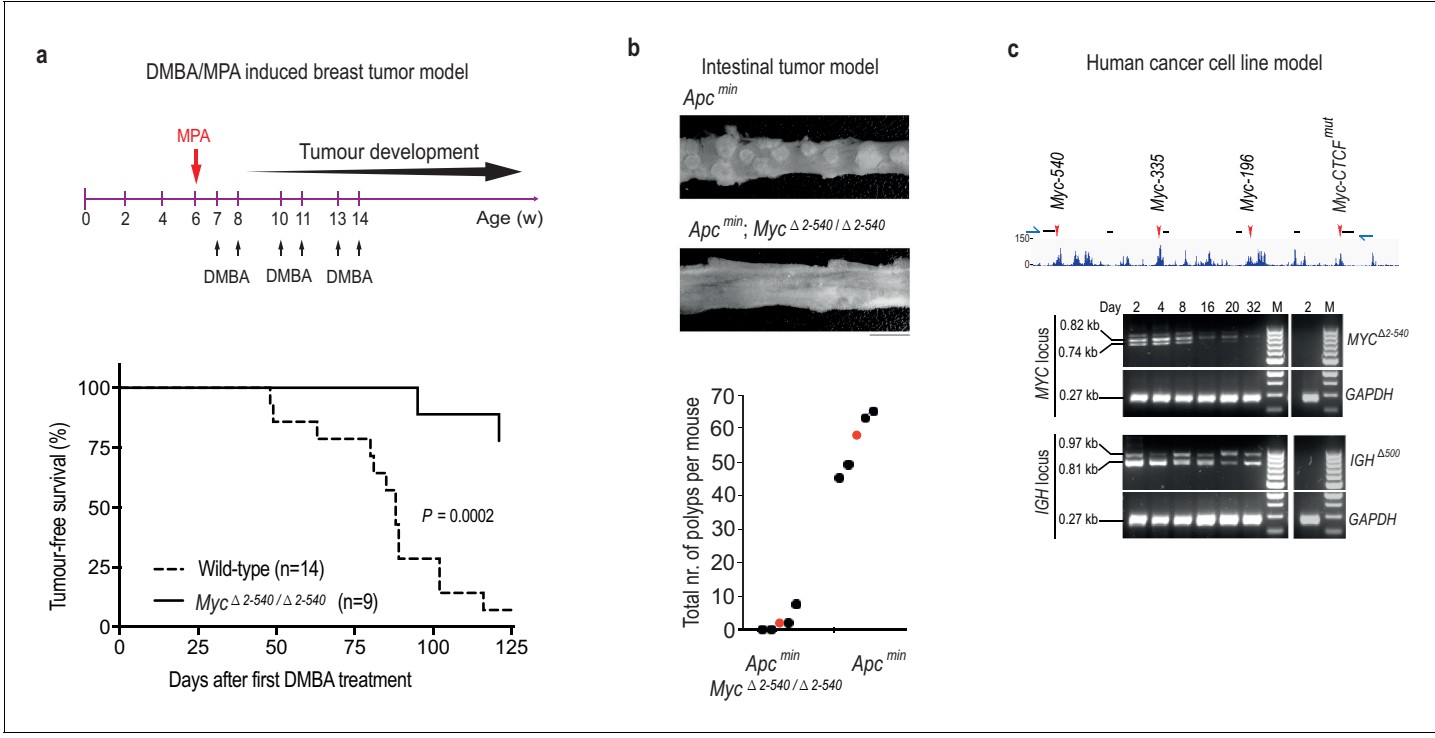

**Figure 6.** *Myc* −2 to −540 kb genomic region is required for the growth of cancers *in vivo* and cancer cells *in vitro*. (a) Tumor-free survival plots showing resistance of $Myc^{\triangle 2-540/\triangle 2-540}$ mice to development of DMBA/MPA induced mammary tumors. p-value=0.0002 (Mantel-Cox Log-rank test). See *Figure 6—source data 1* for details. (b) The *Myc* −2 to −540 kb deletion results in fewer polyps than the *Myc-335* deletion alone. p-value=0.00019 (Students T-test, 2-tailed). $Apc^{min}$ mice were of 4 months of age (n = 5) and $Apc^{min}$; $Myc^{\triangle 2-540/\triangle 2-540}$ mice were 6 months old (n = 5) at the time of analysis. Filled circles correspond to individual mice and red color denotes the median. See *Figure 6—source data 1* for details. Bar equals 5 mm. (c) Crispr-Cas9 mediated deletion of region corresponding to $Myc^{\triangle 2-540/\triangle 2-540}$ in human GP5d colon cancer cells, results in a loss of the edited cells over time. Top panel shows the active enhancer elements in GP5d cells within this region as determined by ChIP-seq analysis of histone H3 lysine 27 acetylation (H3K27ac). The sites of sgRNAs (black lines) and genotyping primers (blue arrows) used are indicated (not to scale). Red arrows mark the enhancer regions used in this study. Bottom panel shows the PCR-genotyping of the *MYC* locus and the control *IGH* locus showing the specific loss of the cells with the edited *MYC* locus over time. *GAPDH* was used as internal control. The right panel in each set shows absence of any deletion in the non-transfected cells (day 2). 100 bp ladder DNA molecular weight marker is shown (M).

The following source data is available for figure 6:

**Source data 1.** Survival time and intestinal polyp numbers for mice in *Figure 6a-b*.

$Apc^{min}$ background. In this study the wild-type mice had on an average 56 polyps at around 4 months of age (n = 5) when they had to be euthanized for ethical reasons similar to what we reported previously. The $Apc^{min}$; $Myc^{\triangle 2-540/\triangle 2-540}$ looked healthy and had on an average 2.4 polyps even at 6 months of age (n = 5) compared to an average of 14.33 polyps reported for the $Apc^{min}$; $Myc$-$335^{-/-}$ null mice at 4 months of age (*Figure 6b*). Together with our earlier findings, these results indicate that loss of the 8q24 super-enhancer region makes mice resistant to both genetically and chemically induced tumors.

We further tested the requirement of this region for the proliferation of cancer cell lines in cultures. We found that the corresponding region (hg19: chr8:128226490–128746456) was also required for GP5d colon cancer cell growth, as indicated by progressive loss of cells bearing a CRISPR/Cas9 induced deletion of the region during co-culture with unedited cells in the population (*Figure 6c*).

## Discussion

The region around the *MYC* gene carries inherited risk towards multiple major forms of cancer. On aggregate, this region contributes more to inherited cancer than any other locus in the human genome. The risk alleles for different cancer types are located in multiple distinct linkage disequilibrium blocks, indicating that different variants contribute to different cancer types. Several of these regions containing risk variants have been implicated in regulation of MYC expression (*Hallikas et al., 2006*; *Sur et al., 2012b*; *Herranz et al., 2014*; *Uslu et al., 2014*), suggesting that a large number of enhancers within this region can drive tumorigenesis. Some of the identified elements have also been shown to have roles in normal development (*Herranz et al., 2014*; *Uslu et al., 2014*).

To study the role of the 8q24 region more systematically, we have in this work deleted several individual enhancer elements, and also analyzed the effect of larger deletions on normal development and carcinogenesis in mice. Our analysis of mice lacking a 538 kb region upstream of the *Myc* gene suggests that enhancer elements within this region cooperatively enhance *Myc* expression. Deletion of individual enhancers in this region has only a weak (*Sur et al., 2012b*) or no effect on *Myc* expression in the mouse intestine in contrast to the deletion of the entire super-enhancer region, which leads to severe decrease in *Myc* expression in multiple tissues.

MYC deficient mouse embryos die due to placental defect at E9.5. The embryos are also smaller in size than wild-type embryos (*Davis et al., 1993*). However, when *Myc* is deleted only in the epiblast, the embryos grow normally and survive until E11.5, when they die due to defects in hematopoiesis (*Dubois et al., 2008*). None of these defects are observed in mice homozygous for the deletion of super-enhancer region. The 8q24 super-enhancer region is thus dispensable for MYC function both in the placenta and during early hematopoiesis. In our mouse colony, the super-enhancer region deficient mice also do not display the size or weight differences reported for *Myc* heterozygous mice that have a 50% reduction in MYC activity (*Trumpp et al., 2001*). These results indicate that tissue-specific enhancers that reside outside of the deleted regions drive sufficient MYC expression in the tissues that contribute to the phenotypes observed in $Myc^{+/-}$ and $Myc^{-/-}$ mice. Consistently with this, several hematopoietic enhancers have been identified in the region 3' of the *MYC* ORF (*Hnisz et al., 2013*; *Shi et al., 2013*; *Herranz et al., 2014*).

*Myc* heterozygous mice also display increased longevity and enhanced healthspan (*Hofmann et al., 2015*). Although the deletion of the super-enhancer region that contains tissue-specific enhancers regulating MYC expression is not equivalent to a heterozygous deletion of the *Myc* gene in the whole body, the $Myc^{\triangle 2-540/\triangle 2-540}$ mice could be an interesting model for identification of the tissues that contribute to the longevity phenotype.

Despite decreased levels of MYC in multiple adult tissues, the mice lacking the super-enhancer region are viable, fertile and display normal tissue morphology in all the tissues we investigated. They display no overt phenotype and do not have marked defects in cell proliferation. The mice are, however, resistant to intestinal tumorigenesis, and DMBA-induced mammary tumors, indicating that this region is important for tumorigenesis also in mice. Our data thus shows that despite the central role of this region in tumorigenesis (*Sur et al., 2012b*; *Lovén et al., 2013*), it is dispensable for normal tissue development and homeostasis under laboratory conditions. Whereas this result may appear very surprising, it is consistent with the original identification of this region using genome-wide association studies (GWAS). GWAS has a high power to identify common variants, and most variants that are common have only a limited effect on physiological functions. This is because a variant that has strong positive or negative effect is rapidly fixed or lost, respectively. Thus, GWAS are specifically biased to find variants that have a relatively large effect on disease, but a small effect on fitness.

Most genes in mammals do not have haploinsufficient phenotypes. Such buffering could be due to mechanisms that maintain constant expression level irrespective of gene dose. However, a simpler buffering mechanism involves either expressing a gene at a very low level where it has no effect, or at a high level where it can contribute its functions even if its expression level is decreased due to transcriptional noise or loss-of-function of one allele. A similar two state mechanism where physiological transcription factor (TF) activity levels in the relevant cell types are either too low to drive any target genes (off state), or high enough to activate all important targets (on state) could also mechanistically explain why most heterozygous null mutations of TF genes have no apparent phenotype.

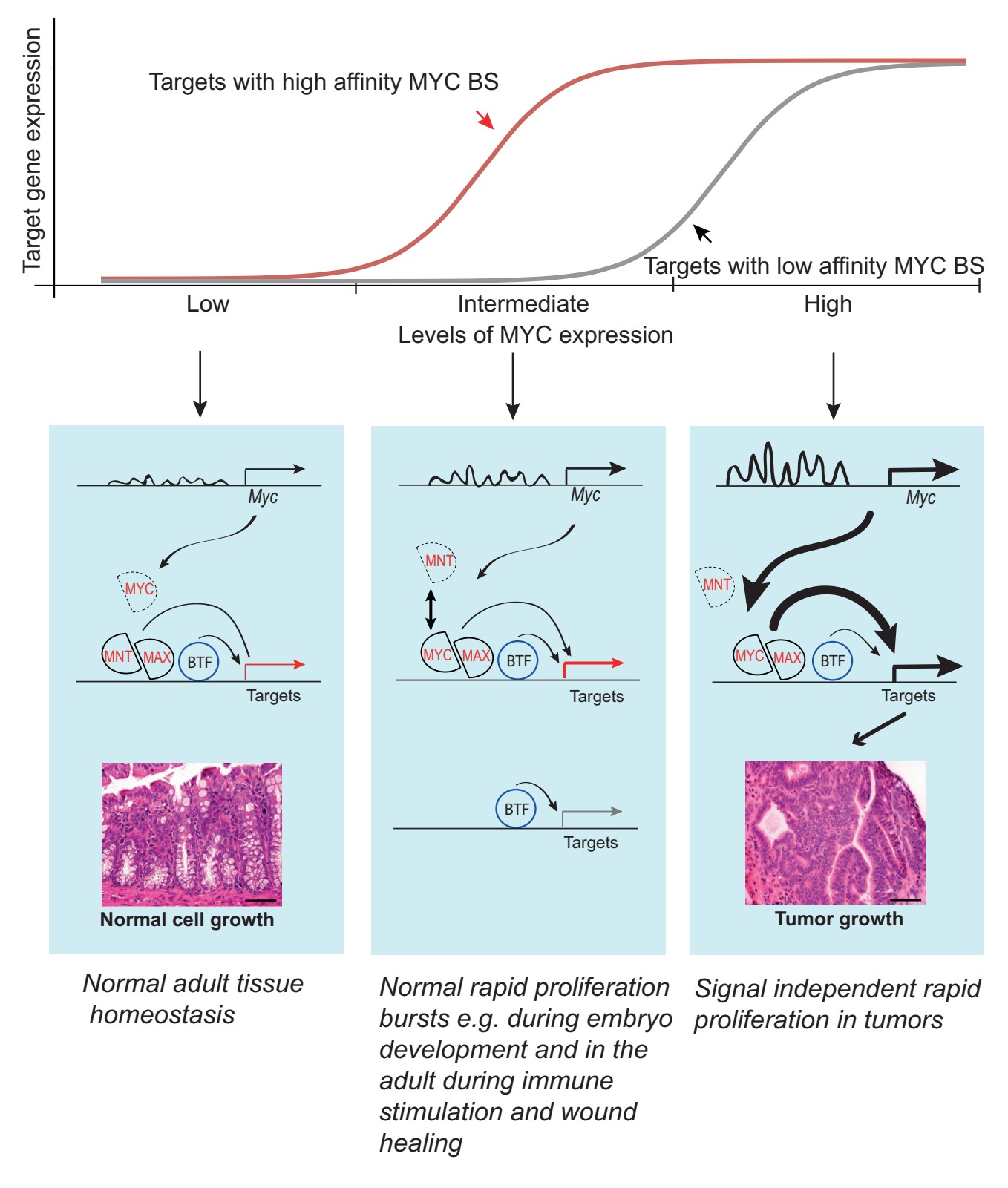

**Figure 7.** Model showing the activity of the *Myc* super-enhancer region during normal homeostasis (left) and cancer (right). During normal tissue homeostasis (left), *Myc* enhancers are not strongly active, and MYC activity is relatively low. The MYC expression level is insufficient to recruit enough MAX proteins to MYC/MAX heterodimers to drive strong induction of the MYC target genes, which instead remain under the control of basal transcription factors (BTF). Under conditions of normal rapid proliferation as seen during embryonic development or during pathological insults in the

*Figure 7 continued on next page*

*Figure 7 continued*

adult, MYC is expressed at intermediate levels to elicit response from targets with high affinity binding sites (red). In cancer cells or cells grown in culture (right), upstream regulators such as Tcf7l2 and β-catenin activate the *Myc* super-enhancers, driving high levels of MYC expression. This leads to the formation of MYC/MAX heterodimers that strongly activate transcription of MYC target genes driving cancer cell growth. The high levels of MYC are also sufficient to induce target genes that harbor low affinity MYC binding sites (grey). The model is consistent with the model of Lorenzin *et al* (*Lorenzin et al., 2016*) who showed genes differ in their response to MYC levels due to differences in the MYC affinity of their promoters. Given that *Myc* super-enhancer region is tumor-specific, and induction of the MYC target genes are not required for normal homeostasis, it provides a promising target for antineoplastic therapies.

Our analysis of the role of MYC in normal colon is consistent with such a simple buffering model (*Figure 7*). However, it should be noted that this buffering mechanism does not operate in all tissues and under all conditions. For example, *Myc* gene dose has effects on mouse size, longevity and hematopoiesis (*Davis et al., 1993*; *Trumpp et al., 2001*; *Dubois et al., 2008*; *Hofmann et al., 2015*). In addition, the level of expression of the *Myc* gene has quantitative effects on cell proliferation under pathological conditions such as activation of T-cells (*Heinzel et al., 2017*). These results indicate that in some situations, MYC is expressed at a level where cell growth responds linearly to small changes in MYC levels (*Figure 7*, middle panel). However, the lack of an overt phenotype in our model under normal physiological conditions in the absence of infection or tissue damage suggests that growth during normal tissue homeostasis in at least some adult tissues does not linearly respond to changes in MYC levels. The lack of an overt phenotype should not, however, be taken to mean that the mice have no phenotype at all. As the super-enhancer region contains several highly conserved DNA segments, and affects cell growth in culture, we expect that it will also affect responses to tissue damage or some other perturbation that we have not investigated here. Therefore, further studies are needed to determine the role of the super-enhancer region in various chronic and acute models of injury and infection.

Based on our data and the earlier literature we propose that under normal physiological conditions in the intestine, the *Myc* gene regulatory system is in the off state, and a basal level of expression of the MYC target genes is maintained by a MYC-independent mechanism. The target genes are thus only sensitive to an increase in MYC levels. Consistently, an 80% decrease of *Myc* mRNA expression does not lead to a proliferation defect, or major changes in expression of known MYC target genes. In contrast, in tumors the system is locked to an on state, where MYC targets are driven to a maximal level by MYC, and the targets are now only sensitive to a decrease in MYC activity (*Figure 7*).

The requirement of MYC in tumor cells appears absolute. In transgenic animal models, overexpression of MYC leads to deregulated proliferation and tumor development in multiple tissues (*Felsher and Bishop, 1999*; *Pelengaris et al., 1999*; *D'Cruz et al., 2001*; *Jain et al., 2002*; *Shachaf et al., 2004*). Furthermore, inhibition of MYC almost invariably causes growth arrest of cancer cells both in culture and *in vivo* (*Soucek et al., 2002*, *2004*; *Hart et al., 2014*). Despite the importance of MYC for cancer growth, it appears that the role of MYC in controlling growth during adult tissue homeostasis is limited. In the adult tissues, MYC is expressed in rapidly proliferating compartments of the body like the intestinal crypts and skin. Deletion of *Myc* in these compartments does not result in prominent proliferation defects (*Wilson et al., 2004*; *Baena et al., 2005*; *Bettess et al., 2005*; *Muncan et al., 2006*). Although there is still controversy regarding MYC requirement for the intestinal homeostasis, in the skin MYC is dispensable under normal adult proliferation and homeostasis *in vivo* (*Oskarsson et al., 2006*). It is however required for Ras mediated tumorigenesis and growth of fibroblasts and keratinocytes *in vitro* (*Mateyak et al., 1997*; *Oskarsson et al., 2006*). Taken together, these results suggest that MYC is required for pathological proliferation, but is less important and in many cases dispensable for normal homeostasis of tissues in the adult. Our results are consistent with these observations.

Prior to our study it was not clear whether the MYC dependence of cancer cells *in vivo* and normal cells in culture is due to shared regulatory mechanisms. Our results have uncovered striking mechanistic similarities between growth of normal cells in culture, and growth of cancer cells *in vivo* by showing that MYC expression depend on the same genetic elements in cultured normal cells and in cancer cells. The similarity between tumor cells and cultured normal cells also suggest that many

potential drugs that block cancer cell growth may have been inadvertently discarded due to their negative effects on growth of normal cells in culture, even when they might not have affected normal tissue homeostasis *in vivo*.

Our results show that the MYC super-enhancer region that carries multi-cancer susceptibility in humans contributes to the formation of multiple tumor types also in mice. Despite its role in tumor formation, it is dispensable for normal development and homeostasis. Loss of the super-enhancer region leads to low MYC expression, but the lowered expression does not translate to changes in expression of MYC target genes in the intestine. Thus, the MYC/MAX/MNT system (*Grandori et al., 2000*) that drives cell growth and proliferation is robustly set to an off state during normal homeostasis, whereas in cancer, the system is locked to a pathological on state. This also explains how physiological growth control can be robust to small perturbations and transcriptional noise. Taken together, our results reveal an important difference between the transcriptional states of normal and cancer cells, and suggest that therapeutic interventions that decrease the activity of the *Myc* super-enhancer region would be well tolerated.

## Materials and methods

### Mouse strains

We generated cKO *Myc-196* and cKO *Myc-540* strains with loxP sites flanking the regions chr15:61445326–61447611 and chr15:61789274–61791107, respectively (Taconic). These mice were crossed to *EIIa-cre* mouse strain (Jackson Laboratory) to generate mice with enhancer deletions. *Myc-CTCF^mut* mouse strain was generated by mutating the CTCF-binding site at chr15:61983375–61983647 TGGCCAGTAGAGGGCAC to TGGAACGTCTTGAATGC. In order to generate large deletions at the *Myc* locus (*Myc^△2-367* and *Myc^△2–540*) *Myc-367⁻* and *Myc-540⁻* were crossed to *Myc-CTCF^mut* that were also heterozygous for the *Rosa26-Cre* (Taconic). The *Myc-540⁻*, *Myc-196⁻* and *Myc-CTCF^mut* carry one lox P site at the respective loci (chr15:61445326, chr15:61618287 and chr15:61983375). The loxP site on chr15:61983375 is located immediately 5' of the mutant CTCF binding site. We obtained compound heterozygotes carrying the chr15:61445326 or the chr15:61618287 loxP site together with the loxP site on chr15:61983375 and the *Rosa26-Cre*. The compound heterozygotes were screened by PCR for the interallelic recombination and the resultant deletion and duplication of the intervening sequence. Mice mosaic for the deletion and duplication were backcrossed to the C57Bl/6 mice in order to segregate the chromosomes carrying the deletion. The F1 heterozygotes were intercrossed to generate mice with homozygous large deletions. *Myc-335* strain has been previously described (*Sur et al., 2012b*). All mice used in the study were on a C57Bl/6 genetic background. All mouse experiments were conducted in accordance with the local ethical guidelines, after approval of the protocols by the ethics committee of the Board of Agriculture, Experimental Animal Authority, Stockholm South, Sweden (Dnr S50/13, S11/15 and S16/15). The sequences of the different primer pairs used for genotypings are given in *Supplementary file 2*.

### Mammary gland whole mount analysis

Inguinal mammary glands were removed from 8 week old virgin females and spread on glass slides. These were fixed for 4 hr in Carnoy's fixative and subsequently stained O/N with Carmine Alum. The whole mounts were rinsed and dehydrated through increasing series of ethanol and cleared in xylene before mounting with the pertex mounting medium.

### Quantitative PCR analysis

qPCR was performed as described previously (*Sur et al., 2012b*). Essentially, total RNA was isolated from whole tissue by homogenizing in RNA Bee reagent (ambios AMS Biotechnology) followed by RNA isolation using Qiagen's RNA MinElute kit according to manufacturers' protocols. 0.5–1 µg of total RNA was reverse transcribed using high capacity reverse transcription kit in a 20 µl reaction (Applied Biosystems). Quantitative PCR in triplicates was performed using the SYBR select master mix (Applied Biosystems) on the LightCycler 480 instrument (Roche). For normalization, mouse *β*-actin transcripts were used as internal controls. Following primer pairs were used for quantitative PCR analysis.

Myc-Fw: 5'-GGGGCTTTGCCTCCGAGCCT-3', Myc-Rev: 5'-TGAGGGGCATC GTCGTGGCT-3', β-actin-Fw: 5'CTGTCGAGTCGCGTCCACCCG-3', β-actin-Rev: 5'-CATGCCGGAGCCGTTGTCGAC-3'.

## RNA-sequencing

NEBNext Ultra Directional RNA library Prep kit (NEB) was used for preparing the samples for RNA-seq together with the NEBNext Poly(A) mRNA magnetic isolation module (NEB) according to manufacturers protocol. In the case of tissues 1–2 µg and for cultured fibroblasts 200 ng of total RNA was used as starting material. For library preparation, adapters and index primers from NEBNext Multiplex Oligos for Illumina kit were used. The RNA-seq library was sequenced on a HiSeq2000 (Illumina). Sequencing reads were mapped to the mouse reference genome (NCBI37/mm9) using Tophat2 (version 2.0.13; RRID:SCR_013035) (*Kim et al., 2013*). Cuffdiff (version 2.2.1; RRID:SCR_001647) was used for differential gene expression analysis and for graphical representation, CummeRbund package (version 2.8.2; RRID:SCR_014568) (*Trapnell et al., 2012*) was used. The upstream regulator analysis was performed on all the significant differentially expressed genes (Cuffdiff q-value <0.05) using QIAGEN's Ingenuity Pathway Analysis (IPA, QIAGEN Redwood city, www.qiagen.com/ingenuity; version 24718999, updated 2015-09-14; RRID:SCR_008653).

## ChIP-seq

ChIP-seq was performed as described in (*Sur et al., 2012b*; *Yan et al., 2013*) with the following modifications: Adult 8–10 week old mice were euthanized and colon was removed, rinsed with cold PBS and cut into fine pieces. Tissue was crosslinked with 1.5% formaldehyde and cultured cells were crosslinked with 1% formaldehyde for 10 min at room temperature and quenched with 0.33M Glycine. Sequences were mapped to the mouse reference genome (NCBI37/mm9) and human reference genome (hg19) using Burrows-Wheeler Alignment tool (bwa) (version 0.6.2) (*Li and Durbin, 2009*) with default parameters. All antibodies used in ChIP-seq experiments were ChIP-grade. In each experiment a non-specific IgG was used as control. Following antibodies were used for ChIP-seq experiments: rabbit anti-H3 lysine 27 acetylation (H3K27ac) (abcam, ab4729: RRID:AB_2118291), mouse anti-H3 lysine four trimethylation (H3K4me3) (abcam, ab1012; RRID:AB_442796), rabbit anti-Rad21 (Santa Cruz, sc-98784; RRID:AB_2238151), goat anti-CTCF (Santa Cruz, sc-15914X; RRID:AB_2086899), rabbit anti-SMC1A (Bethyl Laboratories, A300-055A; RRID:AB_2192467), rabbit IgG (Santa Cruz, sc-2027; RRID:AB_737197), mouse IgG (Santa Cruz, sc-2025; RRID:AB_737182), goat IgG (Santa Cruz, sc-2028; RRID:AB_737167). ChIPseq data for Tcf7l2 was used from ENA accession number PRJEB3354 (*Sur et al., 2012a*) and for GP5d cells from ENA accession number PRJEB1429 (*Yan et al., 2013a*). For visualization, ChIP-seq read depth data were average smoothed across windows of 10 pixels (H3K27ac and H3K4me3) or five pixels (Tcf7l2) in UCSC Genome Browser; RRID: SCR_005780 or alternatively visualized in Integrative Genomics Viewer (IGV, version 2.3; RRID:SCR_011793).

## Bisulfite sequencing

Genomic DNA was isolated using Qiagen's Blood & Tissue Genomic DNA extraction kit. Around 1 µg of wild-type and 250 ng of Myc$^{\triangle2–540/\triangle2\text{-}540}$ null fibroblast genomic DNA was sonicated to 300 bp fragments using Covaris S220 sonicator. Subsequent to end polishing and A base addition, cytosine methylated paired end adapters (Integrated DNA technologies) were ligated to the DNA fragments. The adapter sequences are as follows

    5'-P-GATCGGAAGAGCGGTTCAGCAGGAATGCCGAG
    5'-ACACTCTTTCCCTACACGACGCTCTTCCGATCT

After adapter ligation 300–600 bp fragments were size-selected on a 2% agarose gel. Bisulfite-conversion was carried out using ZYMO EZ DNA Methylation-Gold kit (cat. no. D5005). PCR amplification with 12 and 18 cycles was carried out to prepare libraries from the wild-type and *Myc*$^{\triangle2–540/\triangle2–540}$ null mouse fibroblasts, respectively. The primer pair used for PCR amplification was as follows

PE PCR Primer P1:
5'-AATGATACGGCGACCACCGAGATCTACACTCTTTCCCTACACGACGCTC
    TTCCGATCT

PE PCR Primer P2:

5'-CAAGCAGAAGACGGCATACGAGATCGGTCTCGGCATTCCTGCTGAACC
GCTCTTCCGATCT

The final library was size-selected for 250–300 bp fragments on a 2% agarose gel and 150 bp sequenced from both ends on two lanes of a HiSeq 4000 (Illumina). Raw sequencing reads were quality and adapter trimmed with cutadapt version 1.8.1 (RRID:SCR_011841) in Trim Galore version 0.4.0 (RRID:SCR_011847). Trimming of low-quality ends was done using Phred score cutoff 30. In addition, all reads were trimmed by 2 bp from their 3' end. Adapter trimming was performed using the first 13 bp of the standard Illumina paired-end adapters with stringency overlap two and error rate 0.1. Read alignment was performed against mouse genome mm9 with Bismark (version v0.14.3; RRID:SCR_005604) (*Krueger and Andrews, 2011*) and Bowtie 2 (version 2.2.4; RRID:SCR_005476) (*Langmead and Salzberg, 2012*). Duplicates were removed using the Bismark deduplicate function. Extraction of methylation calls was done with Bismark methylation extractor discarding first 10 bp of both reads and reading methylation calls of overlapping parts of the paired reads from the first read (–no_overlap parameter). Genomic sites with the coverage of at least 10 reads were considered and methylation ratios smoothed with loess method across 49 bp windows.

All sequencing data is uploaded to European Nucleotide Archive (ENA, EMBL-EBI; RRID:SCR_ 006515) under accession number PRJEB11397 (*Dave et al., 2016*; http://www.ebi.ac.uk/ena/data/ view/PRJEB11397).

## Immunohistochemistry and flow cytometry

Five micron paraffin embedded tissue sections were processed for immuno-histochemistry as previously described (*Sur et al., 2012b*). Rabbit polyclonal anti-Myc (Santa Cruz, sc-764; RRID:AB_ 631276) (1:500), Rabbit monoclonal anti Ki-67 (abcam, ab16667; RRID:AB_302459) (1:200), Goat polyclonal anti-Vimentin (Santa Cruz, sc-7557; RRID:AB_793998) (1:500), biotinylated goat anti-Rabbit IgG (Vector Laboratories, BA1000; RRID:AB_2313606) and biotinylated rabbit anti-Goat IgG (Vector Laboratories, BA5000; RRID:AB_2336126) (1:350) antibodies were used. For flow cytometry, single cell suspensions of spleen and bone-marrow and cells from peripheral blood were stained with Fc-block (CD16/CD32 clone 93, Biolegend, 101302, RRID:AB_312801) and subsequently with CD19 (clone 1D3, BD Biosciences, RRID:AB_11154223), TER119 (clone TER119, Biolegend 116210, RRID:AB_313711), CD3ε (clone 145–2 C11, Biolegend 100308, RRID:AB_312673), NK1.1(clone PK136, Biolegend, 108716, RRID:AB_493590), GR1/LY6G (clone RB6-8C5, Biolegend, 108410, RRID: AB_313375), CD4 (clone RM4-5, BD Biosciences, 563747) and CD8a (clone 53–6.7, BD Biosciences, 563332). Dead cells were visualized using Propidium iodide. Samples were analyzed using a BD LSRFortessa instrument.

## Isolation and culture of mouse primary fibroblasts

Fibroblasts were isolated from adult mice by dissecting the skin to ~1 mm$^3$ pieces, and allowing the pieces to adhere to cell culture plates, followed by addition of DMEM medium supplemented with 10% FCS and antibiotics. The fibroblasts were allowed to migrate out from the explants, after which the cells were collected by trypsinization and passaged in the same media for 1–3 passages. For growth assays, $2 \times 10^3$ cells were plated per well in 96 well plates. Cells were trypsinized and counted using hemocytometer at respective time points.

## Tumor induction

### Mammary tumors

Six week-old female mice were implanted s.c. with medroxypregesterone acetate (MPA) pellets (50 mg with a 90 days release period from Innovative Research of America). Subsequently 100 µl of 10 mg/ml dimethylbenz[a]anthracene (DMBA)/oil solution (Sigma) was administered via gavage at 7, 8, 10, 11, 13 and 14 weeks of age. Mice were checked twice a week for development of palpable tumors. Detection of palpable mass in the mammary gland was taken as the end point for tumor-free survival analysis.

## Bladder tumors

Ten week-old male mice were administered 0.1% N-Butyl-N-(4-hydroxybutyl) nitrosamine (BBN) (Sigma) in drinking water for five months. At the end of the treatment the mice were sacrificed and the bladders scored for tumor development.

## Intestinal tumors

$Apc^{min}$ mouse strain (Jackson Laboratory RRID:MGI:5438590) was used as a model for spontaneous development of intestinal tumors.

## CRISPR-Cas9 mediated deletion of super-enhancer region in GP5d cell line

CRISPR-Cas9 mediated deletion of *MYC* super-enhancer region on chromosome 8q24 (GRCh37/hg19 chr8: 128226403–128746490) and Immunoglobulin Heavy (*IGH*) gene locus on chromosome 14q32.33 (GRCh37/hg19 chr14: 106527004–107035452) were carried out in GP5d (Sigma, 95090715; RRID:CVCL_1235, confirmed by STR profiling at ECACC) colon cancer cell line stably expressing Cas9 protein. A lentiviral plasmid containing Cas9 fused via a self-cleaving 2A peptide to a blasticidin resistance gene, was packaged into lentiviral particles using the packaging plasmids psPAX2 (a gift from Didier Trono, Addgene plasmid # 12260, RRID:SCR_002037) and pCMV-VSV-G (a gift from Robert Weinberg (Addgene plasmid # 8454, RRID:SCR_002037). The virus was used to transduce GP5d colon cancer cells. 48 hr after transduction, GP5d cells expressing Cas9 (GP5d-Cas9) were selected in 5 µg/ml Blasticidin (Thermo Fisher Scientific Inc., Cat. no. A1113903). The single guide RNA (sgRNA's) were designed (http://www.broadinstitute.org/rnai/public/analysis-tools/sgrna-design) to span the entire *MYC* super-enhancer region and *IGH* locus (*Figure 6*), respectively (Eurofins MWG Operon). sgRNAs were cloned into an sgRNA Cloning Vector (Addgene Plasmid #41824, RRID:SCR_002037) using Gibson assembly master mix (NEBuilder HiFi DNA assembly Master Mix, Cat no. E2621S). GP5d-Cas9 ($2 \times 10^6$) cells were transfected (using FuGENE HD Transfection Reagent, Cat.no E2312) with 10 µg of eight pooled equimolar sgRNA constructs. Post transfection half of the cultured cells were collected for PCR genotyping, while the other half was re-plated for culturing. Cells were collected at day 2, 4 and subsequently every fourth day till day 32. DNA from cells was extracted (using DNeasy Blood & Tissue Kit, Qiagen Cat. no. 69506) and genotyped with 300 ng of DNA at following conditions - Initial denaturation of 95°C for 5 min; denaturation of 98°C for 15 s, annealing at 60°C for 30 s, extension at 72°C for 30 s (30 cycles for *MYC* super-enhancer region and 35 cycles for *IGH* gene locus deletion genotyping); final extension at 72°C, 5 min. Each experiment was done in triplicate. The sequences of the different guide RNAs and primer pairs used for PCR genotyping of the deletions are given in *Supplementary file 2*. GP5d cells were cultured in DMEM supplemented with 10% FBS and antibiotics. The cell line was mycoplasma free.

## Acknowledgements

We thank Drs. Minna Taipale and Bernhard Schmierer for critical review of the manuscript, and Maria Hoh, Lijuan Hu, Agneta Andersson and Tarja Schröder for technical assistance. We also thank Björn Rozell and Raoul Kuiper, Morphological phenotype analysis core facility at Laboratory Medicine, Karolinska Institute for help with the sectioning and analysis of tissues. This work was supported by the Knut and Alice Wallenberg Foundation (KAW 2013.0088), Center for Innovative Medicine at Karolinska Institutet, and the EU FP7 collaborative project SYSCOL (HEALTH-F5-2010-258236).

## Additional information

### Funding

| Funder | Grant reference number | Author |
| --- | --- | --- |
| Knut och Alice Wallenbergs Stiftelse | KAW 2013.0088 | Jussi Taipale |

| Center for Innovative Medicine at Karolinska Institutet | | Jussi Taipale |
| EU FP7 collaborative project SYSCOL | HEALTH-F5-2010-258236 | Jussi Taipale |

The funders had no role in study design, data collection and interpretation, or the decision to submit the work for publication.

## Author contributions

KD, Data curation, Software, Formal analysis, Validation, Investigation, Visualization, Methodology, Writing—original draft, Project administration, Writing—review and editing; IS, Data curation, Software, Formal analysis, Supervision, Validation, Investigation, Visualization, Methodology, Writing—original draft, Writing—review and editing; JY, LB, Investigation, Methodology, Writing—review and editing; JZ, EK, FZ, Software, Formal analysis, Visualization, Writing—review and editing; XL, SK, CG, ADP, Methodology, Writing—review and editing; RM, Formal analysis, Investigation, Methodology, Writing—review and editing; JT, Conceptualization, Resources, Formal analysis, Supervision, Funding acquisition, Writing—original draft, Project administration, Writing—review and editing

## Author ORCIDs

Kashyap Dave, http://orcid.org/0000-0003-0332-8362
Inderpreet Sur, http://orcid.org/0000-0001-5787-1040
Jussi Taipale, http://orcid.org/0000-0003-4204-0951

## Ethics

Animal experimentation: All mouse experiments were conducted in accordance with the local ethical guidelines after approval of the protocols by the ethics committee of the The Board of Agriculture, Experimental Animal Authority, Stockholm South, Sweden (Dnr. S50/13, S11/15 and S16/15).

# Additional files

## Supplementary files

• Supplementary file 1. List of genes that show significant differential expression in the wild-type and the $Myc^{\triangle 2-540/\triangle 2-540}$ colon (q-value <0.05).

• Supplementary file 2. List of primers and guide RNA sequences used in this study.

## Major datasets

The following datasets were generated:

| Author(s) | Year | Dataset title | Dataset URL | Database, license, and accessibility information |
| --- | --- | --- | --- | --- |
| Dave K, Sur I, Yan J, Zhang J, Kaasinen E, Zhong F, Blaas L, Li X, Kharazi S, Gustafson C, De Paepe A, Månsson R, Taipale J | 2016 | Mice deficient of Myc super-enhancer region reveal differential control mechanism between normal and pathological growth | http://www.ebi.ac.uk/ena/data/view/PRJEB11397 | Publicly available at the EMBL-EBI (accession no: PRJEB11397) |
| Dave K, Sur I, Yan J, Zhang J, Kaasinen E, Zhong F, Blaas L, Li X, Kharazi S, Gustafson C, De Paepe A, Månsson R and Taipale J | 2017 | Mapping of promoter enhancer usage and interaction in hematopoietic development | http://www.ebi.ac.uk/ena/data/view/PRJEB20316 | Publicly available at the EMBL-EBI (accession no: PRJEB20316) |

The following previously published datasets were used:

| Author(s) | Year | Dataset title | Dataset URL | Database, license, and accessibility information |
|---|---|---|---|---|
| Sur IK, Hallikas O, Vähärautio A, Yan J, Turunen M, Enge M, Taipale M, Karhu A, Aaltonen LA, Taipale J | 2012 | Mice Lacking a Myc Enhancer Element that Includes Human SNP rs6983267 Are Resistant to Intestinal Tumors. | http://www.ebi.ac.uk/ena/data/view/PRJEB3354 | Publicly available at the EMBL-EBI (accession no: PRJEB3354) |
| Yan J., Enge M., Whitington T., Dave K., Liu J., Sur I., Schmierer B., Jolma A., Kivioja T., Taipale M., Taipale J | 2013 | Transcription factor binding in human cells occurs in dense clusters formed around cohesin anchor sites | http://www.ebi.ac.uk/ena/data/view/PRJEB1429 | Publicly available at the EMBL-EBI (accession no: PRJEB1429) |

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

## Appendix 1

### Purification of primary mouse cells

Bone marrow (BM) cells harvested from 8- to 12 weeks old C57BL/6 mice. For isolation of LSK Flt3⁻ cells, BM cells were subjected to depletion of mature cells using a cocktail of purified antibodies containing TER119 (TER-119; Biolegend 116202; RRID:AB_313703), CD19 (1D3; BD, 553783; RRID:AB_395047), CD3 (17A2; BD, 555273; RRID:AB_395697), GR1 (RB6-8C5; Biolegend, 108402; RRID:AB_313367) and CD11b (M1/70; Biolegend, 101202; RRID:AB_312785) in combination with sheep anti-rat IgG Dynabeads (Invitrogen; 11035). Non-depleted cells were stained with Ter119 PECy5 (TER-119; Biolegend, 116210; RRID:AB_313711), NK1.1 PECy5 (PK136; Biolegend, 108716; RRID:AB_493590), CD3 PECy5 (145–2 C11; Biolegend, 100310; RRID:AB_312675), GR1 PECy5 (RB6-8C5; Biolegend, 108410; RRID:AB_313375), CD11b PECy5 (M1/70; Biolegend, 101210; RRID:AB_312793), CD19 PECF594 (1D3; BD, 562291; RRID:AB_11154223), KIT (CD117) APCeFlour780 (2B8; eBioscience, 47-1171-82; RRID:AB_1272177), SCA1 PB (D7; Biolegend, 108120; RRID:AB_493273), FLT3 (CD135) PE (A2F10; Biolegend, 135306; RRID:AB_1877217), CD11C PECy7 (N418; Biolegend, 117318; RRID:AB_493568), LY6C APC (HK1.4: Biolegend, 128016; RRID:AB_1732076) and IL7R (CD127) biotin (A7R34 Biolegend 135006; RRID:AB_2126118; visualized using Streptavidin-QD655; Invitrogen Q10121MP). LSK FLT3⁻ (HSCs) cells were subsequently FACS sorted as TER119/CD3/GR1/NK1.1/ MAC1$^{low/-}$CD19⁻LY6C⁻CD11C⁻IL7R⁻SCA1⁺KIT⁺FLT3⁻. For isolation of mature B cells, BM cells were subjected to MACS column enrichment of CD45R (B220)+ cells using anti-CD45R beads (Miltenyi Biotec, 130-049-501). B220 enriched cells were stained with TER119 PECy5 (TER-119), GR1 PECy5 (RB6-8C5), CD11b PECy5 (M1/70), IgD PB (11–26c.2a; Biolegend, 405712; RRID:AB_1937244), IgM PECy7 (11/14; eBioscience 25-5790-82; RRID:AB_469655), CD19 PECF594 (1D3). Mature B cells subsequently FACS sorted as TER119/GR1/ MAC1⁻CD19⁺IgM⁺IgD⁺. Propidium iodide (Life technologies, p3566) was used as a dead cell discriminator when sorting live cells (for RNAseq experiments) and Aqua fluorescent reactive dye (Life technologies; L34957) when sorting fixed cells (for ChIPseq experiments).

For ChIPseq experiments, fully antibody/viability dye-stained cells ($5 \times 10^6$ cells/ml) were fixed by incubation with 1% formaldehyde (ThermoFisher Scientific; 28908) for 10 min at room temperature (RT). Formaldehyde was quenched using 0.1 vol 1 M glycine and incubated for 10 min at RT. Cells were additionally washed with 0.1 M glycine before being resuspended in PBS with 2% FCS prior to FACS sorting. Cell sorting was done on a BD FACSAriaIII cell sorter (BD Biosciences).

### ChIP-sequencing

3 µg of polyclonal anti-H3K27Ac (Diagenode, cat# C15410196, lot# A1723-0041D, RRID:AB_2637079) or H3K4me2 (Millipore, cat#07–030, lot#2089140 and lot#2309072, RRID:AB_11213050) antibody was bound to 10 µl Protein G-coupled Dynabeads (ThermoFisher) per ChIP and incubated with rotation for 4 hr at 4°C. Pellets of $0.5 \times 10^6$ PFA-fixed cells were resuspended in 100 µl SDS lysis buffer (50 mM Tris/HCl, 0.5% SDS, and 10 mM EDTA), placed cold for 15 min and sonicated for 12 cycles of 30 s on/30 s off on high power using a Bioruptor Plus (Diagenode). Samples were centrifuged, and supernatants transferred to new tubes. After addition of 200 µl of ChIP dilution buffer (50 mM Tris/HCl, 225 mM NaCl, 0.15% NaDoc, and 1.5% Triton-X) and 4 µl of 50X protease inhibitors (Roche), samples were incubated at room temperature for 10 min. 10% of each sample was saved for input controls. Antibody-coated dynabeads were washed, resuspended with cell lysate and rotated overnight at 4°C.

Immunoprecipitated chromatin was washed once with low salt buffer (50 mM Tris/HCl, 150 mM NaCl, 0.1% SDS, 0.1% NaDOC, 1% Triton X-100, 1 mM EDTA), high salt buffer (50 mM Tris/HCl, 500 mM NaCl, 0.1% SDS, 0.1% NaDoc, 1% Triton X-100, and 1 mM EDTA) and LiCl buffer (10 mM Tris/HCl, 250 mM LiCl, 0.5% IGEPAL CA-630, 0.5% NaDoc, 1 mM EDTA) followed by two washings with TE buffer. For reversal of crosslinking, chromatin complexes and input control samples were diluted in 200 μl ChIP elution buffer (10 mM Tris/HCl, 0.5% SDS, 300 mM NaCl, and 5 mM EDTA) and 2 μl of 20 μg/ml proteinase K (Thermo Scientific). Samples were vortexed and incubated shaking overnight at 65°C. After reverse crosslinking, 1 μl 20 μg/ml RNAse (Sigma) was added and incubated at 37°C for 30 min. After another 2 hr of incubation with 2 μl of proteinase K at 55°C, samples were placed in a magnet to trap magnetic beads and supernatant collected. DNA purification was carried out using Qiagen MinElute PCR Purification Kit.

DNA concentrations in purified samples were measured using Qubit dsDNA HS Kit (Invitrogen). Libraries were prepared using Rubicon ThruPLEX DNA-seq 12S Kit, according to manufacturer's instructions. 2 ng of chromatin was used when available but samples below Qubit detection levels (<0.5–1.5 ng) were frequently used. After 11 cycles of PCR amplification, adapter cleanup was done using Agencourt AmPureXP beads (Beckman Coulter) at a ratio of 1:0.88. Libraries with an average size of 400–500 bp were pooled and single-end sequenced (50 cycles) using the Illumina sequencing platform (HiSeq2000).

## RNA-sequencing

For RNA extraction 5,000–10,000 cells were sorted into buffer RLT (Qiagen) with $\beta$-mercaptoethanol and total RNA was extracted using Rneasy Micro Kit (Qiagen) according to manufacturers instructions. On-column DNase I treatment was performed to minimize DNA contamination. Strand specific RNAseq libraries were prepared using TotalScript RNA-seq kit (Epicentre) according to the manufacturer's instructions. Barcoded libraries were pooled and pair-end sequenced (2 × 50 cycles) using the Illumina platform (mainly HighSeq 2500).

## ChIP-seq data analysis

Quality of sequencing samples was assessed with FastQC (v0.11.2). Samples were mapped to the mm10 genome using Bowtie2 (v2.2.3) with default parameters (*Langmead and Salzberg, 2012*). Mapped reads were filtered with HOMER (v4.6) (*Heinz et al., 2010*) using the makeTagDirectory command, only keeping uniquely mapped reads and removing possible PCR duplicates by restricting the tags per base pair to 1 (-tbp 1). Resulting filtered reads were visualized by generating bigWig files from tag directories, using HOMER's makeBigWig.pl with a set fragment length (fragLength 130) and normalization to 10 million reads (-norm 1e7).

## RNA-seq data analysis

Quality of sequencing samples was assessed with FastQC (v0.11.2) (*Andrews, 2010*). Samples were mapped to the mm10 genome using STAR (v2.4) with default parameters for paired-end reads (*Dobin et al., 2013*). Mapped reads were filtered with HOMER (v4.6) using the makeTagDirectory command with strand specific pair-end read settings (-sspe), and removing excessive possible PCR duplicates by restricting the tags per base pair to 3 (-tbp 3). Resulting filtered reads were visualized by generating bigWig files from tag directories, using HOMER's makeBigWig.pl with a set fragment length (-fragLength 75), normalization to 10 million reads (-norm 1e7) and stranded data setting (-strand).

The above sequencing data is accessible via ENA accession number PRJEB20316 (*Dave et al., 2017*; http://www.ebi.ac.uk/ena/data/view/PRJEB20316).

