## [Decision Letter]

Thank you for submitting your article "Mice deficient of *Myc* super-enhancer region reveal differential control mechanism between normal and pathological growth" for consideration by *eLife*. Your article has been favorably evaluated by Sean Morrison (Senior Editor) and two reviewers, one of whom, Chi Van Dang, is a member of our Board of Reviewing Editors.

The reviewers have significant concerns about your current manuscript, discussed the reviews with one another, and the Reviewing Editor has drafted this decision to help you prepare your responses to determine whether a formal revised submission will be invited.

Summary:

Dave et al. reports the effect of deleting 538kb enhancer (termed "super-enhancer) region upstream of murine Myc on development and tumorigenesis. This region contains susceptibility loci with several SNPs previously correlated with increased risk for human prostate and colorectal cancer. The authors had previously shown that alteration of the risk allele G of rs6983267 in a mouse model modifies risk for intestinal tumorigenesis, but did not affect viability. Multiple risk alleles have been identified in the enhancer (super-enhancer) region, and the authors addressed the consequence of deleting this region on normal mouse development. They document that deletion of a 538kb region reduced normal *Myc* expression by up to 80% and yet mutant mice were viable without overt phenotype. Importantly, tumorigenesis (e.g., DMBA-induced mammary tumors) is severely diminished in this mutant background, and cells isolated from these mice have slowed growth. Overall, this manuscript contains instructive observations about the regulation of *Myc* expression, physiology and pathophysiological expression of *Myc* for tumorigenesis, underscoring the role of deregulated MYC expression in human cancers.

Essential revisions:

1) The authors construed from these observations that "the MYC/MAX/MNT system that drives cell growth and proliferation is robustly set to an off state in normal homeostasis, whereas in cancer, the system is locked into a pathological on state." This statement in the last paragraph of the Discussion section is not supported by the evidence, given that the authors have not fully studied normal cellular physiology. *Myc* expression under normal physiological conditions could vary according to signaling and hence the concept of an 'on versus off' state seems over-simplified. For example, Heinzel et al. (Nature Immunol 18:96-104 (2017); see reference 17 below) demonstrated that *Myc* expression dose effect is influenced by input signals into T cell activation. In this regard, the authors should document whether normal T cell activation is altered in the mutant *Myc* animals.

2) The *Myc* heterozygotes (Sedivy and coworkers: Hofmann et al. Cell 2015) have increased longevity. It stands to reason that the *Myc* super-enhancer mutant may also have increased longevity due to decreased *Myc* expression. In this regard, the manuscript could be further enriched by examining whether mTOR signaling is decreased in the super-enhancer *Myc* mutant tissues or cells.

3) The authors build upon their previous work in which they showed elegantly that a 1.7 kb region containing a SNP in a TCF binding site 335 kb upstream of the MYC TSS is associated with slightly higher levels of *Myc*, and that these slightly higher *Myc* levels considerably enhance tumorigenesis. The reduced susceptibility to tumorigenesis of the 2-540 deletion is an interesting observation, but it would be nice to include some sort of assurance that the effects are not mediated solely through the 335 region that has already been reported and characterized by the authors (and others) and shown confer resistance to tumor formation when deleted in *Apc^min^* mice. Are DMBA/MPA induced mammary tumors influenced by the 335-region deletion or are separate segments of the 2-540 region involved?

4) The authors are a bit loose with the term super-enhancer (1, 2). The labeling and characterization of the 2-540 kb deleted region as a super-enhancer, especially under physiological conditions is quite questionable. All of the literature on *Myc* and its surrounding gene desert indicates that super-enhancer commissioning seems to be a pathological and not physiological event, often associated with translocations that juxtapose super-enhancers from far-distant regions, usually on other chromosomes (3-5). It seems to be a bit premature to define 0.5 megabases as a super-enhancer. This is quite large, even for a super-enhancer and the authors have performed no functional experiments or computational analysis to discriminate this region as a super-enhancer versus a collection of typical enhancers that each may act independently to mediate the actions of different signaling systems and factors (for example estrogen) (6). The only indication in human cells that there is a potential native super-enhancer in this region that may be commissioned in pancreatic cancer or colon cancer is that of Hnisz et al., but data in these same papers suggest that this region is not normally a super-enhancer.

5) The increased expression of *Myc* in splenic B-cells deleted of 2-540 suggests that this region does not act a purely as an enhancer, but rather that there must be tissue specific negative cis-region(s) interposed amongst the positive acting sequences. The authors should examine chromatin marks, DHS sites etc. as well as the 2-367 B-cells to see if they might learn anything about the nature of the B-cell repressive region. In any case, these results argue against this being a huge, monotonically positive acting super-enhancer.

6) Does the whole "super-enhancer" region act as a single unit or do different regions show a proclivity to act in different tissues? Do the 2-540 and 2-367 deletions behave similarly with respect to tumorigenesis and to carcinogens across different tissues?

7) The authors state that they see no effect of the 2-540 deletion on expression other in tumors and in splenic B-cells. The authors should study carefully the recent *eLife* paper by Lorenzin et al. (8) and then reexamine their own Figure 5. Lorenzin et al. report that maneuvers that decrease MYC lead to preferential decrease in expression of the most highly expressed genes that are most easily saturated with MYC. Note that in Figure 5 left-colon that the blue dots almost all fall below the diagonal of identity for the most highly expressed genes just as would be expected if *Myc* acts according to Lorenzin. Lorenzin et al. also report that as the promoters of the most highly expressed genes increases further, the excess MYC progressively backfills the promoters of more weakly expressed genes thus augmenting the output of middle level expressed genes. Note that in Figure 5 left spleen, that the green dots bulge above the diagonal in the middle region, again just as explained by Lorenzin et al. and in accord with the amplifier model of MYC action. Thus, it seems that low levels of *Myc* are not inert, but that the cellular systems are robust enough to handle the changes in *Myc* under these conditions.

8) The authors assert that there are no phenotypic consequences of lowering *Myc*. They cite the paper of Dubois et al. (8) that *Myc* expression is only essential in the placenta, but not in the epiblast (embryo proper) except when the embryos die a couple of days later at E11.5 from a failure of hematopoietic stem cells; when the embryos die all other tissues appeared normal for their stage of development-however almost half of gestation remained to be completed and there is nothing in the Dubois paper to indicate that growth and maturation of all remaining tissues proceeds normally through gestation and after birth. Many studies have found phenotypes associated with decreased levels of *Myc*. Trumpp et al. (9) constructed an allelic series that clearly showed that habitus size changes according to the amount of *Myc*. In light of this, the authors need to conduct some careful measurements of animal and organ weight, size and cell numbers to substantiate their claim that there is no phenotype. A superficial perusal of the thoracic and abdominal contents might miss subtle changes. What about limb length-limb size that has been reported to be influenced by *Myc*?

9) Very significantly, the authors' conclusions about *Myc* acting digitally off-on by weak versus strong expression to buffer its effects, is at odds with a large and significant literature that says that small changes in MYC are sometimes very physiologically significant apart from tumors, and that even very low levels of MYC may be important (for example fibroblasts may survive with low MYC, but they do not survive without MYC (see the work of Sedivy, etc.). It is also at odds with reports describing haploinsufficiency of *Myc*. Unlike normal fibroblasts, fibroblasts haploinsufficient for MYC cannot be immortalized by expression of hTERT (10). Mice haploinsufficient for *Myc* live longer and are smaller (9, 11). While *Myc* is dispensable for hepatocytes (12, 13), a bit of *Myc* seems to be required for colonic crypt cells (14). Less than two-fold differences in MYC expression (up or down) provoke super-competition in which the cells with more MYC kill and replace the cells with less MYC (in flies and in mice) (15, 16). Slight differential changes in *Myc* modify the temporal profile of the immune response (17). There are many more examples. Small differences in MYC seem to be associated with many, sometimes subtle, manifestations making it difficult to accept the digital interpretation proposed by the authors.

1) Pott S, Lieb JD: What are super-enhancers? Nature genetics 2015, 47:8-12.

2) Whyte WA, Orlando DA, Hnisz D, Abraham BJ, Lin CY, Kagey MH, Rahl PB, Lee TI, Young RA: Master transcription factors and mediator establish super-enhancers at key cell identity genes. Cell 2013, 153:307-19.

3) Hnisz D, Abraham BJ, Lee TI, Lau A, Saint-Andre V, Sigova AA, Hoke HA, Young RA: Super-enhancers in the control of cell identity and disease. Cell 2013, 155:934-47.

4) Hnisz D, Schuijers J, Lin CY, Weintraub AS, Abraham BJ, Lee TI, Bradner JE, Young RA: Convergence of developmental and oncogenic signaling pathways at transcriptional super-enhancers. Molecular cell 2015, 58:362-70.

5) Loven J, Hoke HA, Lin CY, Lau A, Orlando DA, Vakoc CR, Bradner JE, Lee TI, Young RA: Selective inhibition of tumor oncogenes by disruption of super-enhancers. Cell 2013, 153:320-34.

6) Wang C, Mayer JA, Mazumdar A, Fertuck K, Kim H, Brown M, Brown PH: Estrogen induces c-myc gene expression via an upstream enhancer activated by the estrogen receptor and the AP-1 transcription factor. Molecular endocrinology (Baltimore, Md) 2011, 25:1527-38.

7) Lorenzin F, Benary U, Baluapuri A, Walz S, Jung LA, von Eyss B, Kisker C, Wolf J, Eilers M, Wolf E: Different promoter affinities account for specificity in MYC-dependent gene regulation. *eLife* 2016, 5.

8) Dubois NC, Adolphe C, Ehninger A, Wang RA, Robertson EJ, Trumpp A: Placental rescue reveals a sole requirement for c-*Myc* in embryonic erythroblast survival and hematopoietic stem cell function. Development (Cambridge, England) 2008, 135:2455-65.

9) Trumpp A, Refaeli Y, Oskarsson T, Gasser S, Murphy M, Martin GR, Bishop JM: c-*Myc* regulates mammalian body size by controlling cell number but not cell size. Nature 2001, 414:768-73.

10) Guney I, Wu S, Sedivy JM: Reduced c-*Myc* signaling triggers telomere-independent senescence by regulating Bmi-1 and p16(INK4a). Proceedings of the National Academy of Sciences of the United States of America 2006, 103:3645-50.

11) Hofmann JW, Zhao X, De Cecco M, Peterson AL, Pagliaroli L, Manivannan J, Hubbard GB, Ikeno Y, Zhang Y, Feng B, Li X, Serre T, Qi W, Van Remmen H, Miller RA, Bath KG, de Cabo R, Xu H, Neretti N, Sedivy JM: Reduced expression of MYC increases longevity and enhances healthspan. Cell 2015, 160:477-88.

12) Sanders JA, Schorl C, Patel A, Sedivy JM, Gruppuso PA: Postnatal liver growth and regeneration are independent of c-myc in a mouse model of conditional hepatic c-myc deletion. BMC physiology 2012, 12:1.

13) Li F, Xiang Y, Potter J, Dinavahi R, Dang CV, Lee LA: Conditional deletion of c-myc does not impair liver regeneration. Cancer research 2006, 66:5608-12.

14) Muncan V, Sansom OJ, Tertoolen L, Phesse TJ, Begthel H, Sancho E, Cole AM, Gregorieff A, de Alboran IM, Clevers H, Clarke AR: Rapid loss of intestinal crypts upon conditional deletion of the Wnt/Tcf-4 target gene c-*Myc*. Molecular and cellular biology 2006, 26:8418-26.

15) Claveria C, Giovinazzo G, Sierra R, Torres M: *Myc*-driven endogenous cell competition in the early mammalian embryo. Nature 2013, 500:39-44.

16) Johnston LA: Socializing with MYC: cell competition in development and as a model for premalignant cancer. Cold Spring Harbor perspectives in medicine 2014, 4:a014274.

17) Heinzel S, Binh Giang T, Kan A, Marchingo JM, Lye BK, Corcoran LM, Hodgkin PD: A *Myc*-dependent division timer complements a cell-death timer to regulate T cell and B cell responses. Nature immunology 2017, 18:96-103.

---

## [Author Response]

*Essential revisions:*

*1) The authors construed from these observations that "the MYC/MAX/MNT system that drives cell growth and proliferation is robustly set to an off state in normal homeostasis, whereas in cancer, the system is locked into a pathological on state." This statement in the last paragraph of the Discussion section is not supported by the evidence, given that the authors have not fully studied normal cellular physiology. Myc expression under normal physiological conditions could vary according to signaling and hence the concept of an 'on versus off' state seems over-simplified. For example, Heinzel et al. (Nature Immunol 18:96-104 (2017); see reference 17 below) demonstrated that Myc expression dose effect is influenced by input signals into T cell activation. In this regard, the authors should document whether normal T cell activation is altered in the mutant Myc animals.*

We agree with the reviewer that *Myc* is important in immune cells, but would like to maintain a distinction between normal homeostasis and pathological states in the manuscript. We have included additional data in Figure 2—figure supplement 1 showing the lack of a genotype-dependent effect on T-cell number in these mice. We also do not expect T-cell activation to be strongly affected, as enhancers active in T-cells are on the other side of the *Myc* ORF (Herranz et al. 2014), and the mice are healthy and do not display an immunodeficient phenotype. We have now clarified this (subsection “Loss of the super-enhancer region leads to tissue-specific changes in *Myc* expression”, second paragraph).

To further clarify the manuscript, we have also now made it clearer that T-cell activation occurs generally under pathological conditions and does not fit under our definition of "normal cellular homeostasis" (Discussion, sixth paragraph). As the distinction between normal conditions in laboratory mice, and pathological conditions is central to the manuscript, in our view it would still be very important to clarify this point.

We have also now added a paragraph to the Discussion section to discuss the various pathological conditions where *Myc* is clearly involved, and now state that it will be important to assess the role of the 538 kb region in these processes in future studies.

We feel that a detailed study of T-cell activation kinetics is beyond the scope of the current work. However, if such an experiment is essential for publication, we can run the experiment and provide the data. Given that our mouse facility is moving, it would delay publication approximately 6 months.

*2) The Myc heterozygotes (Sedivy and coworkers: Hofmann et al. Cell 2015) have increased longevity. It stands to reason that the Myc super-enhancer mutant may also have increased longevity due to decreased Myc expression. In this regard, the manuscript could be further enriched by examining whether mTOR signaling is decreased in the super-enhancer Myc mutant tissues or cells.*

We agree that the finding of *Myc* contributing to longevity is interesting, and we are aware that Hoffman et al. showed the reduction in age related symptoms in *Myc^+/-^* mice. However, the deletion of a super-enhancer region that contains tissue-specific enhancers regulating *Myc* expression in our model is not equivalent to a heterozygous deletion of the *Myc* gene in the whole mouse body. So we expect that the effect of the enhancer deletion on longevity, if any, to be tissue specific or minor. To detect an effect on longevity would require a large cohort of mice as well as ethical permits to age the mice, which we cannot obtain in Sweden (we have to sacrifice mice that are old or sick). The analysis would also take 2-3 years. Thus, analysis of longevity is not feasible within the scope of the present study. As the effects of the enhancer deletion are tissue-specific, molecular analysis in the absence of phenotypic data would not be informative. To address this point, we have added a sentence to the Discussion section citing the Hofmann et al. paper and pointing out that the mouse could be used as a model to study which tissues contribute to the longevity phenotype (Discussion, fourth paragraph).

*3) The authors build upon their previous work in which they showed elegantly that a 1.7 kb region containing a SNP in a TCF binding site 335 kb upstream of the MYC TSS is associated with slightly higher levels of Myc, and that these slightly higher Myc levels considerably enhance tumorigenesis. The reduced susceptibility to tumorigenesis of the 2-540 deletion is an interesting observation, but it would be nice to include some sort of assurance that the effects are not mediated solely through the 335 region that has already been reported and characterized by the authors (and others) and shown confer resistance to tumor formation when deleted in APC^min^ mice. Are DMBA/MPA induced mammary tumors influenced by the 335-region deletion or are separate segments of the 2-540 region involved?*

We agree that it is important to analyze whether the effects of the large deletion are solely due to the *Myc-335* region. We have now included additional discussion and data to show that this is not the case. The large region deleted in this mouse model contains several tissue-specific enhancers in addition to the *Myc-335* (Tuupanen et al. 2009; Ahmadiyeh et al. 2010); this is clarified in the fourth paragraph of the Introduction. In addition, the breast cancer susceptibility locus in humans is distinct from the colon cancer susceptibility locus that harbors Myc-335, and there is thus no reason to expect that *Myc-335* would contribute to breast cancer. This is now stated in the first paragraph of the subsection “The Myc super-enhancer region is required for tumorigenesis in mice”. Although we cannot rule out the possibility at this time that *Myc-335* contributes to the DMBA/MPA induced mammary tumors, additional elements outside of the *Myc-335* region are definitely also playing a role in tumor models. We have now included data to show that the 2-540 region deletion in an *Apc^min^* background results in fewer polyps than the *Myc-335* deletion alone (new Figure 6). In this study the wild-type mice had on an average 56 polyps at around 4 months of age (n=5) when they had to be euthanized for ethical reasons similar to what we reported previously. The mice with the 2-540 deletion looked healthy and had on an average 2.4 polyps even at 6 months of age (n=5) compared to an average of 14.33 polyps we reported for the *Myc-335* null mice at 4 months of age. These results are consistent with multiple Tcf7l2 ChIP-seq peaks that are observed in the colon within this region (Figure 2). This data is now shown in Figure 6, and the results are discussed in the second paragraph of the subsection “The *Myc* super-enhancer region is required for tumorigenesis in mice”.

*4) The authors are a bit loose with the term super-enhancer (1, 2). The labeling and characterization of the 2-540 kb deleted region as a super-enhancer, especially under physiological conditions is quite questionable. All of the literature on Myc and its surrounding gene desert indicates that super-enhancer commissioning seems to be a pathological and not physiological event, often associated with translocations that juxtapose super-enhancers from far-distant regions, usually on other chromosomes (3-5). It seems to be a bit premature to define 0.5 megabases as a super-enhancer. This is quite large, even for a super-enhancer and the authors have performed no functional experiments or computational analysis to discriminate this region as a super-enhancer versus a collection of typical enhancers that each may act independently to mediate the actions of different signaling systems and factors (for example estrogen) (6). The only indication in human cells that there is a potential native super-enhancer in this region that may be commissioned in pancreatic cancer or colon cancer is that of Hnisz et al., but data in these same papers suggest that this region is not normally a super-enhancer.*

We agree that the term super-enhancer is somewhat controversial, and that the precise boundaries of presumed regulatory regions vary in different cell types and under different conditions. The region deleted in this study contains several cancer susceptibility loci and some of these have been linked to the presence of tissue-specific enhancers active in cancer cells. As the reviewer points out, in most cases the enhancers or super-enhancers identified within this region appear to be very tumor specific (Ahmadiyeh et al. 2010; Hnisz et al. 2013; Loven et al. 2013). This suggests that they regulate *Myc* under pathological condition without affecting the proliferation/ homeostasis of normal cells. Our results confirm this idea. Since this region contains several tissue-specific and cancer-specific regulatory elements that fall both under the definition of enhancers and super-enhancers we have referred to the deleted region as the "super-enhancer region" (not "super-enhancer"). We have now clarified this definition in the Introduction (last paragraph).

*5) The increased expression of Myc in splenic B-cells deleted of 2-540 suggests that this region does not act a purely as an enhancer, but rather that there must be tissue specific negative cis-region(s) interposed amongst the positive acting sequences. The authors should examine chromatin marks, DHS sites etc. as well as the 2-367 B-cells to see if they might learn anything about the nature of the B-cell repressive region. In any case, these results argue against this being a huge, monotonically positive acting super-enhancer.*

We agree that increased *Myc* expression in the splenic B-cells can suggest the presence of a negative cis-region. However, a more likely possibility is that the deletion of 538 kb brings an enhancer outside of this region closer to the *Myc* promoter. This is because a known B-cell specific enhancer at -556 has been described as the target for EBNA2 mediated large-scale directional reorganization of *Myc* promoter-enhancer interactions (Wood et al. 2016). We have now discussed this issue (subsection “Loss of the super-enhancer region leads to tissue-specific changes in *Myc* expression”, third paragraph and in legend for Figure 2—figure supplement 1), and included chromatin analysis of wild-type mouse B-cells in the revised version that show the enhancers in the region that is retained in the deletion (new Figure 2—figure supplement 1).

*6) Does the whole "super-enhancer" region act as a single unit or do different regions show a proclivity to act in different tissues? Do the 2-540 and 2-367 deletions behave similarly with respect to tumorigenesis and to carcinogens across different tissues?*

Several studies have shown that this region contains enhancers that act in a tissue-specific manner and that the different cancer susceptibilities span distinct segments of this region. The data presented in this manuscript also shows that the active enhancer marks span different regions in different tissues/cells (see colon vs. fibroblast ChIP-seq data). We do not think that the whole super-enhancer acts as a single unit, rather specific sub-regions regulate MYC in a tissue-specific manner. We have now clarified this in the revised version (Introduction, last paragraph).

*7) The authors state that they see no effect of the 2-540 deletion on expression other in tumors and in splenic B-cells. The authors should study carefully the recent eLife paper by Lorenzin et al. (7) and then reexamine their own Figure 5. Lorenzin et al. report that maneuvers that decrease MYC lead to preferential decrease in expression of the most highly expressed genes that are most easily saturated with MYC. Note that in Figure 5 left-colon that the blue dots almost all fall below the diagonal of identity for the most highly expressed genes just as would be expected if Myc acts according to Lorenzin. Lorenzin et al. also report that as the promoters of the most highly expressed genes increases further, the excess MYC progressively backfills the promoters of more weakly expressed genes thus augmenting the output of middle level expressed genes. Note that in Figure 5 left spleen, that the green dots bulge above the diagonal in the middle region, again just as explained by Lorenzin et al. and in accord with the amplifier model of MYC action. Thus, it seems that low levels of Myc are not inert, but that the cellular systems are robust enough to handle the changes in Myc under these conditions.*

We agree that in cultured cells *Myc* is active. Lorenzin et al. nicely showed that due to differences in the MYC affinity to promoters, genes differ in their response to MYC levels. We can clarify that the more detailed model proposed by Lorenzin et al. and our model are not mutually exclusive. The `off-state´ of *Myc* under resting physiological conditions in our model is lower than that found in cultured cells by Lorenzin et al. That is why we see little effect on target gene expression due to further downregulation of *Myc*. To address this point we compared wild-type and 2-540 null FPKM values for gene sets from Lorenzin et al. paper in our colon and spleen RNA-seq data. We did not find differences in either the high-affinity binding or the low-affinity binding gene sets. We also plotted the median FPKM values of gene transcripts for the colon data since one of the wild-type samples for unknown reason had higher amounts of ribosomal structural protein transcripts. This is now included as new Figure 5—figure supplement 1.

In addition, we have added an intermediate state to the model, and also discuss the responses of low affinity and high affinity targets (Discussion, sixth paragraph).

*8) The authors assert that there are no phenotypic consequences of lowering Myc. They cite the paper of Dubois et al. (8) that Myc expression is only essential in the placenta, but not in the epiblast (embryo proper) except when the embryos die a couple of days later at E11.5 from a failure of hematopoietic stem cells; when the embryos die all other tissues appeared normal for their stage of development-however almost half of gestation remained to be completed and there is nothing in the Dubois paper to indicate that growth and maturation of all remaining tissues proceeds normally through gestation and after birth. Many studies have found phenotypes associated with decreased levels of Myc. Trumpp et al. (9) constructed an allelic series that clearly showed that habitus size changes according to the amount of Myc. In light of this, the authors need to conduct some careful measurements of animal and organ weight, size and cell numbers to substantiate their claim that there is no phenotype. A superficial perusal of the thoracic and abdominal contents might miss subtle changes. What about limb length-limb size that has been reported to be influenced by Myc?*

We agree that Trumpp et al. correlated a reduced body size and weight with decreased *Myc* levels. However the paper by Duboiset al. found no effect on the size of embryos lacking *Myc* thus attributing the effect on the weight/ body size to the placental requirement of *Myc*. We agree with the reviewer that in the study by Dubois et al. nearly half of the gestation stage is still remaining and could result in smaller weight or size. Heterozygous mice in the study by Trumpp et al. had on an average a reduction of 18-22% in body mass and were visibly distinguishable from the wild-type littermates. We have now clarified that we cannot see visible size differences of 2-540 null mice, and do not detect a significant weight difference in our mouse colony based on 17 wild-type and 15 null mice (Discussion, third paragraph). We have also now explained that since we are deleting very tissue specific enhancers as opposed to the *Myc* gene deletion, we do not expect striking difference on the body weight or size.

Detection of any more subtle changes in body weight or organ size would require a large cohort of mice and even if such effects were found, they would not materially affect the main point of the manuscript. Regarding the tissues we focused on prostate, intestine, spleen, mammary gland and bladder based on the cancer susceptibility loci identified within this region. Histological examination of these tissues does not show morphological defects. We have found that the total number of spleen cells are reduced which could be due to the decreased number of B-cells shown in the manuscript. We have now included this data in the revised version (legend to Figure 2—figure supplement 1). We have also now added an explicit sentence to the Discussion stating that we do not claim that the mice have "no phenotype", but rather that there is "no overt phenotype" that affects the viability or fertility of the mice under laboratory conditions (Discussion, sixth paragraph).

*9) Very significantly, the authors' conclusions about Myc acting digitally off-on by weak versus strong expression to buffer its effects, is at odds with a large and significant literature that says that small changes in MYC are sometimes very physiologically significant apart from tumors, and that even very low levels of MYC may be important (for example fibroblasts may survive with low MYC, but they do not survive without MYC (see the work of Sedivy, etc.). It is also at odds with reports describing haploinsufficiency of Myc. Unlike normal fibroblasts, fibroblasts haploinsufficient for MYC cannot be immortalized by expression of hTERT (10). Mice haploinsufficient for Myc live longer and are smaller (9, 11). While Myc is dispensable for hepatocytes (12, 13), a bit of Myc seems to be required for colonic crypt cells (14). Less than two-fold differences in MYC expression (up or down) provoke super-competition in which the cells with more MYC kill and replace the cells with less MYC (in flies and in mice) (15, 16). Slight differential changes in Myc modify the temporal profile of the immune response (17). There are many more examples. Small differences in MYC seem to be associated with many, sometimes subtle, manifestations making it difficult to accept the digital interpretation proposed by the authors.*

We agree with the reviewer that *Myc* is required for several aspects of the embryonic development in mice, and in the adult hematopoietic system (e.g. (Wilson et al. 2004)) and in cultured cells. *Myc* heterozygous mice are also smaller, from birth to adulthood, presumably due to the effect on placenta or embryonic development. We have now included more discussion of these phenotypes in the revised version (Discussion, third paragraph).

However, several studies have documented that loss of *Myc* appears to have little effect on normal tissue homeostasis of most adult tissues in vivo. In the skin, loss of *Myc* under normal conditions is well tolerated in vivo while keratinocytes with a *Myc* deletion exhibit reduced proliferation in vitro (Oskarsson et al. 2006). In the case of the intestine where *Myc* is expressed in the proliferating crypts, there are conflicting reports. Although Muncan et al. (2006) showed the loss of *Myc* null crypts in their study, a parallel study (Bettess et al. 2005) showed that the proliferation and maintenance of the adult crypts was not dependent on *Myc* expression. To add to the confusion, in Sansom et al. (2007) some of the authors of the Muncan et al. paper cite their own work as evidence of "remarkably little effect of [*Myc*] gene deletion on intestinal proliferation". Thus, it appears that the matter is controversial. We have not detected prominent defects associated with the 2-540 null genotype in the intestine. We think this is not surprising given that proliferation in vivo can occur in the absence / reduced levels of *Myc* (discussed above) and that we have a tissue-specific effect on the *Myc* regulation.

In addition, in our model we do not see any prominent *Myc*-associated phenotypes that have been previously reported e.g. in the haematopoietic system which is consistent with the fact that enhancers regulating the hematological malignancies or normal immune cell function have been identified around 1 Mb downstream of *Myc* gene and not included in our deletions (Herranz et al. 2014).

We have now addressed the literature on *Myc* by citing the relevant references, and pointing out that the results are in some cases conflicting, but overall it is clear that *Myc* is critical in tumors, and required for growth of cultured cells and during some developmental processes (Discussion, third and sixth paragraphs). In our mice, we do not see major effects, which could theoretically be due to the remaining *Myc* levels, but more likely caused by the tissue-specific role of the region that was deleted. This is now stated in the aforementioned paragraph.

We have also modified the model to cite the previous papers and to clarify that *Myc* levels also matter, and that the "on" and "off" states as depicted in the original model do not explain some developmental phenotypes or quantitative regulation of immune cell responses (as in Heinzel et al). We have added an "intermediate *Myc* level" panel to the model to address this, and to clarify that the two extreme states are relevant to "cancer", and to "normal physiological conditions in adult tissue homeostasis in the absence of tissue damage or infection" (Discussion, sixth paragraph and Figure 7).